# Cerebellar modulation of synaptic input to freezing-related neurons in the periaqueductal gray

Christopher E Vaaga, Spencer T Brown, Indira M Raman*

Department of Neurobiology, Northwestern University, Evanston, United States

**Abstract** Innate defensive behaviors, such as freezing, are adaptive for avoiding predation. Freezing-related midbrain regions project to the cerebellum, which is known to regulate rapid sensorimotor integration, raising the question of cerebellar contributions to freezing. Here, we find that neurons of the mouse medial (fastigial) cerebellar nuclei (mCbN), which fire spontaneously with wide dynamic ranges, send glutamatergic projections to the ventrolateral periaqueductal gray (vlPAG), which contains diverse cell types. In freely moving mice, optogenetically stimulating glutamatergic vlPAG neurons that express Chx10 reliably induces freezing. In vlPAG slices, mCbN terminals excite ~20% of neurons positive for Chx10 or GAD2 and ~70% of dopaminergic TH-positive neurons. Stimulating either mCbN afferents or TH neurons augments IPSCs and suppresses EPSCs in Chx10 neurons by activating postsynaptic $D_2$ receptors. The results suggest that mCbN activity regulates dopaminergic modulation of the vlPAG, favoring inhibition of Chx10 neurons. Suppression of cerebellar output may therefore facilitate freezing.

## Introduction

To avoid predation, animals must rapidly recognize and respond to threats. Such defensive behaviors rely on innate neural circuitry both to identify the threatening stimulus within the context of the local environment and to engage one of multiple defensive behaviors, such as freezing or fleeing (*Blanchard and Blanchard, 1972*; *De Franceschi et al., 2016*; *Fendt and Fanselow, 1999*; *LeDoux, 2000*; *Yilmaz and Meister, 2013*), depending on the imminence of the threat (*Perusini and Fanselow, 2015*). At a neuronal level, the specific defensive strategy that is selected is determined by which one of multiple rostro-caudal columns is activated within the periaqueductal gray (*Bandler et al., 2000*; *Bandler and Shipley, 1994*; *Carrive, 1993*; *Keay and Bandler, 2001*; *Koutsikou et al., 2017*; *Watson et al., 2016*). Freezing, for example, depends on the ventrolateral column of the periaqueductal gray (vlPAG), which contains glutamatergic neurons whose activation elicits freezing and whose inactivation blocks non-associative, 'innate' freezing to intrinsically threatening stimuli (*Tovote et al., 2016*). In addition to eliciting fear-related outputs (*Oka et al., 2008*; *Perusini and Fanselow, 2015*; *Tovote et al., 2015*), recent evidence suggests that vlPAG neurons also participate in assessing threat probability (*Wright and McDannald, 2019*). Consistent with its role in integrating complex, contextual sensory stimuli, the vlPAG receives input from many brain areas, several of which participate in conditioned freezing, including the amygdala, hypothalamus, zona incerta and prefrontal cortex (*Tovote et al., 2015*). Remaining questions, however, are how neurons of the vlPAG integrate synaptic inputs, whether those inputs might be subject to short-term neuromodulation, and, if so, where such modulation might arise.

Despite not being widely recognized as a component of fear-related circuitry, the cerebellum has the capacity to influence freezing behavior (*Apps and Strata, 2015*). Anatomically, the medial cerebellar nucleus (mCbN in rodents; the fastigial nucleus in primates), which receives input from the cerebellar vermis, projects to the vlPAG (*Teune et al., 2000*; *Gonzalo-Ruiz et al., 1990*). Although this

*For correspondence:
i-raman@northwestern.edu

Competing interests: The authors declare that no competing interests exist.

projection has been attributed primarily to oculomotor function, lesions of the cerebellar vermis lead to decreases in both innate and conditioned freezing in rodents (*Supple et al., 1988*; *Koutsikou et al., 2014*; *Sacchetti et al., 2002*). Given known roles of the cerebellum in sensorimotor integration, it seems plausible that it may participate in perception of potential threats, prediction of threat probability, and/or execution of innate or conditioned freezing behavior. Indeed, consistent with the well-established role of the cerebellum in associative learning, fear conditioning leads to potentiation of parallel fiber synapses onto vermal Purkinje cells, and deficits in cerebellar plasticity disrupt fear recall (*Sacchetti et al., 2004*). More generally, the cerebellum regulates movement, and freezing is a motor behavior characterized by the suppression of voluntary motion (*Koutsikou et al., 2014*), raising the possibility of a more fundamental cerebellar role in innate freezing.

To investigate the influence of the cerebellum on freezing related circuitry in the vlPAG, we studied a subset of vlPAG neurons whose direct activation drives freezing, examined their intrinsic and synaptic properties, tested for cerebellar input to these cells, and explored the synaptic mechanisms by which cerebellar activity could influence vlPAG output. We identified a population of vlPAG neurons that expresses Chx10 and whose optogenetic stimulation in vivo elicits reliable and robust freezing. Inputs from the mCbN directly excite a subset of these freezing-related neurons and also innervate local dopaminergic neurons in the vlPAG, which in turn modulate the relative strength of electrically evoked EPSCs and IPSCs to favor inhibition. These findings suggest that cerebellar input to the vlPAG may regulate freezing by altering how synaptic signals are integrated within the vlPAG microcircuit.

## Results

### Projections of the medial cerebellar nucleus to the ventrolateral periaqueductal gray

We reasoned that the influence of the cerebellar vermis on innate freezing (*Supple et al., 1988*; *Koutsikou et al., 2014*) might result from direct synaptic connections in the ventrolateral periaqueductal gray. Previous tracing studies have demonstrated that the mCbN indeed projects to the vlPAG (*Gonzalo-Ruiz et al., 1990*; *Teune et al., 2000*), and electrical stimulation of the mCbN elicits short latency field potentials in the vlPAG (*Whiteside and Snider, 1953*), but this projection has historically been thought to contribute to oculomotor function. Therefore, we investigated whether the medial cerebellar nucleus projects specifically to the caudal vlPAG, the site of freezing-related circuitry. First, we injected the mCbN with viruses expressing a channelrhodopsin-eYFP (ChR2-eYFP) fusion protein (*Figure 1A*, *left, middle*). After 4–6 weeks, axonal labeling was evident in the caudal-most aspect of the vlPAG, consistent with a direct projection (*Figure 1A*, *right*). Axonal labeling had the highest density in the caudal ~600–900 μm of the vlPAG, ventral and lateral to the central aqueduct, with sparser axonal labeling near the aqueduct (*Figure 1—figure supplement 1*). Conversely, injecting the vlPAG with either CTb-GFP or retrobeads (*Figure 1B*, *left, middle*) resulted in retrograde labeling of large neurons in the mCbN (*Figure 1B*, *right*). Following a unilateral injection of retrograde tracer, the greatest labeling density was in the contralateral mCbN, although some ipsilateral labeling was also evident. Retrogradely labeled neurons were not observed in the neighboring interpositus or lateral cerebellar nucleus, suggesting specificity of the projection from the mCbN to the vlPAG.

Even the vlPAG, however, is heterogeneous, as pharmacological activation of the vlPAG elicits freezing, bradycardia, and anti-nociception (*Bandler et al., 2000*). A subset of glutamatergic neurons in the vlPAG that elicit freezing without associated analgesia have projections to the magnocellular reticular nucleus (Mc), defined as including the gigantocellularis pars ventralis (GiV) and lateral paragigantocellular nucleus (*Esposito et al., 2014*; *Tovote et al., 2016*); cells in these medullary areas in turn project directly to hindlimb and forelimb motor neurons in the spinal cord (*Esposito et al., 2014*). To identify these vlPAG projection neurons in particular, we took advantage of the fact that freezing can also be elicited by stimulating a subset of glutamatergic neurons in the vlPAG that express Chx10, a homeodomain transcription factor (*Leiras et al., 2017*, SfN abstract). We therefore examined Chx10 neurons in the PAG of Chx10-cre mice expressing tdTomato ('Chx10-

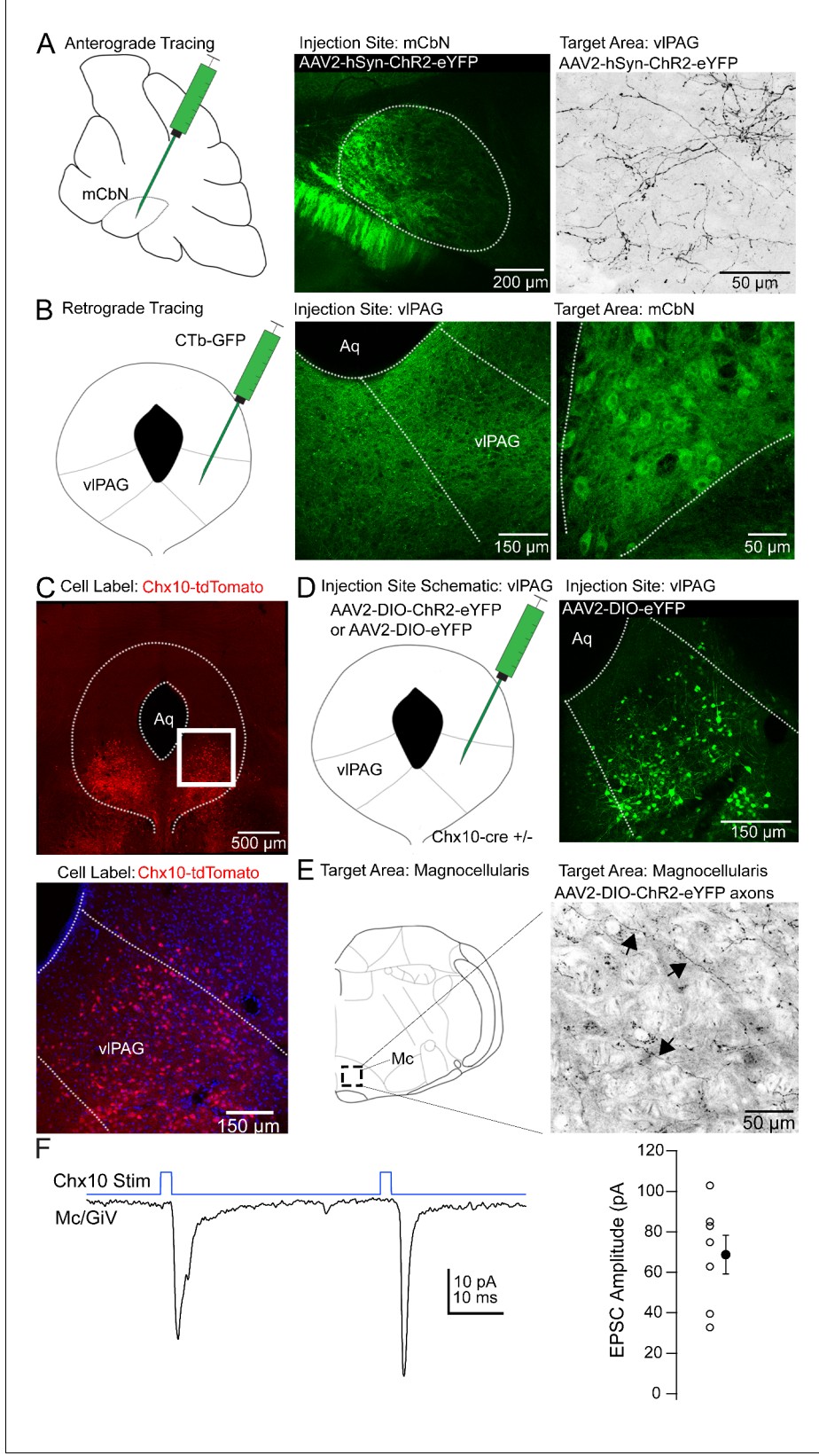

**Figure 1.** Anatomical tracing and identification of Chx10-positive vlPAG neurons. (**A**) *Left,* Schematic of a parasagittal section of the cerebellum showing injection site for anterograde tracing from mCbN. *Middle,* Confocal image of the mCbN after injection of AAV-hSyn-ChR2-eYFP virus. *Dotted line,* mCbN boundaries. *Right, Figure 1 continued on next page*

*Figure 1 continued*

Confocal image of virally labeled mCbN axons in the vlPAG. (B) *Left,* Schematic of a coronal section of the PAG showing injection site for retrograde tracing from the vlPAG. *Middle,* Example injection site of CTb-GFP in the vlPAG. *Dotted line,* approximate boundaries of vlPAG. *Right* Retrogradely labeled neurons in the mCbN. *Dotted line,* boundaries of mCbN. (C) *Top,* Low magnification confocal image of the PAG in a Chx10-tdT mouse, showing the distribution of Chx10 neurons in the ventrolateral PAG. *Bottom,* High magnification image of the *white* box in the upper panel. (D) *Left,* Schematic of the PAG showing injection site for anterograde tracing in Chx10-cre mice. *Right,* Confocal image of the vlPAG after viral labeling of Chx10 neurons. (E) *Left,* Schematic of a coronal section of the brainstem showing the approximate rostro-caudal position of Chx10-positive axons in the magnocellular reticular nucleus (Mc), approximately −6.6 mm from bregma. *Right,* Axonal labeling of Chx10-positive axons in the Mc. Arrows indicate labeled axons. (F) *Left,* EPSCs evoked in Mc neurons by optogenetic stimulation of Chx10-ChR2 axons. *Right,* Population data for first evoked EPSC in Mc neurons. *Open symbols,* individual cells, *solid symbols,* mean ± SEM.

The online version of this article includes the following figure supplement(s) for figure 1:

**Figure supplement 1.** Low magnification image of mCbN labeled axons in vlPAG.

tdT') to test whether these might project to freezing-associated areas of the medulla, with the goal of investigating their sensitivity to cerebellar input.

Chx10 neurons were enriched in the vlPAG (*Figure 1C*), and injecting viruses with cre-dependent transgene expression into the vlPAG of Chx10-cre mice (*Figure 1D*) resulted in relatively selective axonal labeling in the Mc (*Figure 1E*), with few labeled fibers in the more dorsal nucleus gigantocellularis. To assess the probability that Chx10 neurons also participate in analgesia, we looked for projections to anti-nociceptive regions of the rostral ventral medulla (RVM; *Fields and Heinricher, 1985*; *Zorman et al., 1981*). Chx10 labeled axons, however, were primarily caudal to the traditional RVM circuitry, having the highest density at the rostal-caudal level of the inferior olive, making it seem unlikely that these cells participate in the anti-nociceptive pathway from the vlPAG.

To test whether the Chx10 neurons indeed made functional excitatory contacts in the Mc, we made whole-cell recordings from neurons in the Mc in Chx10-ChR2 mice. Light stimulation of Chx10 neurons resulted in inward currents of −68.7 ± 9.6 pA at −70 mV in Mc neurons, consistent with glutamatergic synapses (*Figure 1F*; n = 7 cells [5 M, 2 F]). In the 4 cells in which pairs of responses were evoked (40 ms interval), the paired pulse ratio was 0.7 ± 0.4. Together, these results suggest that Chx10 neurons in the vlPAG have the attributes necessary to influence freezing-related behaviors through an excitatory projection to the Mc.

## Optogenetic stimulation of vlPAG Chx10 neurons in vivo

If Chx10 neurons are indeed the vlPAG neurons that are part of the freezing circuit, then their activation should suppress or limit movement. First, to verify that light stimulation could effectively excite Chx10 neurons for prolonged periods, we recorded in PAG slices from Chx10-ChR2 mice. Indeed, in Chx10 neurons (n = 9 cells, [9M, 0F]), 50 Hz trains of 100 light stimuli (10 ms pulses) elicited photocurrents that were relatively stable under voltage clamp (1st current, −78.5 ± 8.6 pA; 5th and 100th current, 73.1 ± 2.0% and 66.0 ± 1.7% of 1st current). Under current-clamp, the same stimulation brought firing rates from 3.8 ± 3.9 spikes/sec to 20.5 ± 3.6 spikes/sec, with elevated spike rates persisting throughout stimulation (*Figure 2A,B*).

Next, we stimulated Chx10 neurons in the vlPAG in vivo while monitoring the activity of freely-moving mice. To do so, we implanted a unilateral fiber optic cannula above the vlPAG in Chx10-ChR2 mice and control mice with Chx10 labeled but lacking ChR2 (Chx10-tdT) (*Figure 2C,D*). In the nine brains that were recovered, the fiber track confirmed that the cannula had been positioned just dorsal to the vlPAG with mean coordinates relative to bregma: anterior-posterior, −4.5 mm (range: −4.2 to −4.8 mm); medial-lateral, 0.6 mm (range: 0.3 to 0.8 mm); dorsal-ventral, −2.6 mm (range: −2.8 to −2.25 mm). Light trains as in slices (10 ms pulses at 50 Hz) applied for 2–5 s resulted in a nearly complete cessation of movement in Chx10-ChR2 mice, which persisted for the duration of optogenetic stimulation (n = 7 mice, 50 trials per mouse; *Figure 2E*, *left*). Visual inspection of the behavior was consistent with freezing, as mice ceased all voluntary movements with the exception of respiration, eye movements, and some whisking (*Video 1*). Freezing was elicited regardless of the ongoing motor behavior of the mouse, and was rarely accompanied by threat assessment behavior

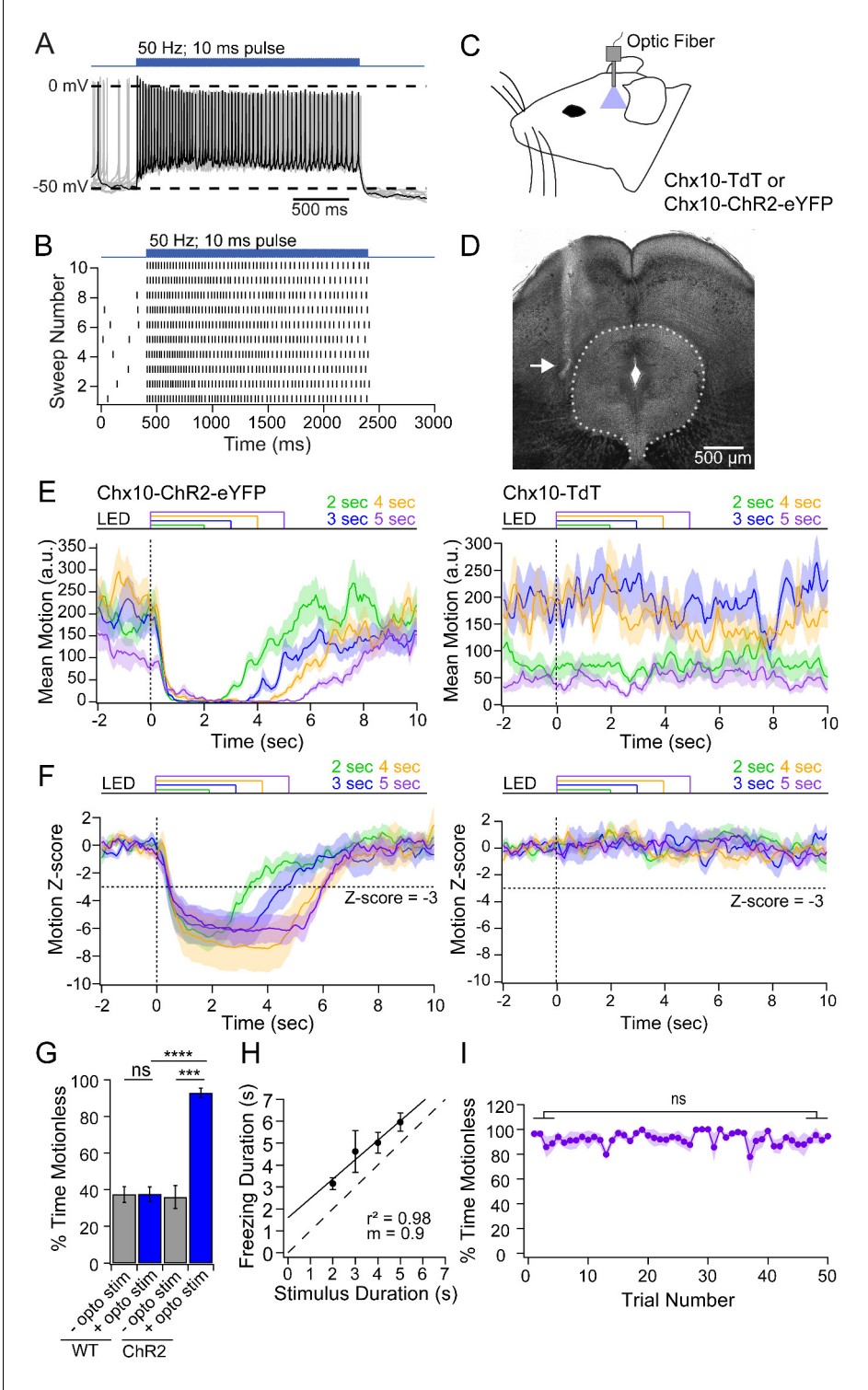

**Figure 2.** Freezing evoked by activation of Chx10 neurons in vivo. (**A**) Action potentials in a Chx10-ChR2 expressing neuron in the vlPAG during a train (100 stimuli, 50 Hz, 10 ms) of light. (**B**) Spike raster of 10 consecutive sweeps during stimulation of the cell in A. (**C**) Schematic of a mouse showing placement of the fiber optic cannula in the vlPAG. (**D**) Transmitted-light image of the midbrain showing placement of fiber optic cannula just lateral to the vlPAG (*white arrow*). (**E**) Plot of the mean motion as a function of time in Chx10-ChR2 mice (*left*) and Chx10-tdT mice (*right*). Time 0 indicates onset of LED optogenetic stimulation (*dotted line*). Each trace is the mean of 50 consecutive sweeps in a single mouse. (**F**) Z-score of mean motion for each stimulus duration for all Chx10-ChR2

*Figure 2 continued on next page*

*Figure 2 continued*

mice (*left*) and Chx10-tdT mice *right*). *Horizontal dotted line*, a Z-score of −3, *vertical dotted line*, light onset. (**G**) Mean percent time immobile with and without stimulation in Chx10-TdT mice and Chx10-ChR2. (**H**) Freezing duration *vs.* the stimulus duration in Chx10-ChR2 mice. *Solid symbols*, mean ± SEM. *Dashed line*, unity. (**I**) Stability of mean percent time freezing in response to 5 s optogenetic stimulations across 50 consecutive trials in Chx10-ChR2 mice.

or continued freezing after stimulus termination, consistent with the idea that Chx10 vlPAG activation was not aversive but directly evoked a motor program. Conversely, in control mice (n = 7 mice), optogenetic stimuli had no detectable effect on ongoing movement (*Figure 2E*, *right*).

The data across mice were quantified by the z-score of the movement index. Relative to the mean baseline movement for 2 s before optogenetic stimulation, movement fell by >3 standard deviations (SDs) in Chx10-ChR2 mice but was unchanged in control mice (*Figure 2F*). Owing to the sampling rate of 3.75 frames/sec, the latency to freezing onset could not be determined precisely. We therefore measured the latency to a 3-SD drop, which was 660 ± 30 ms (~2.5 frames). The stability of freezing after the onset of immobility was therefore quantified as the percent of time below the 3-SD threshold starting 1 s after stimulus onset. In the pre-stimulus baseline, immobility was comparable in Chx10-ChR2 and control mice (35.9 ± 6.2% and 37.4 ± 4.2%, respectively, n = 7 mice per group). During stimulation, immobility increased to 92.8 ± 2.5% in Chx10-ChR2 mice (p=<0.001, paired t-test) but remained at 37.6 ± 3.9% in control mice (p=0.84, paired t-test; Chx10-ChR2 *vs.* control, p=<0.0001, unpaired t-test, *Figure 2G*). The duration of immobility and of light stimulation were strongly correlated (*Figure 2H*, $r^2$ = 0.98), with a slope near unity (0.9) suggesting that freezing and Chx10 neuronal firing are directly related. Finally, consistent with a role in directly evoking the freezing motor pattern, repeated optogenetic stimulation (50 trials, 20 s inter-trial interval) of Chx10 neurons reliably elicited freezing, without habituation (*Figure 2I*; stimuli 1–5: 92.2 ± 2.2%; stimuli 46–50: 92.0 ± 1.3%, p=0.9, unpaired t-test). Together, these data provide evidence that vlPAG Chx10 neurons directly excite medullary neurons that evoke motor programs associated with freezing.

## Intrinsic and synaptic properties of Chx10 vlPAG neurons

To understand the firing patterns by which vlPAG Chx10 cells may drive freezing behavior, we examined their intrinsic and synaptic properties in slices of the vlPAG (*Table 1*). Chx10 neurons were electrically tight, with input resistances of 584.1 ± 47.3 MΩ and capacitances of 24.2 ± 1.3 pF (n = 26 cells [15 M, 11 F]). Current-clamped Chx10 neurons fired spontaneously at 5.8 ± 1.2 spikes/s (*Figure 3A,B*; n = 28 cells). In contrast GAD2+ neurons recorded in the vlPAG fired more rapidly, at 20.7 ± 3.2 spikes/sec (*Figure 3—figure supplement 1*; n = 16 cells [8 M, 8 F], Chx10 *vs.* GAD2, p=0.0003; unpaired t-test). Chx10 and GAD2 neurons also differed in their action potential waveforms (*Figure 3C,D*). In Chx10 neurons, spikes were broad (half-width: 1.0 ± 0.05 ms, n = 26 cells [15 M, 11 F]) and lacked afterhyperpolarizations, whereas those of GAD2+ vlPAG neurons were briefer, with prominent afterhyperpolarizations (*Figure 3—figure supplement 1*). The differences support the idea that Chx10 neurons are glutamatergic.

Hyperpolarizing and depolarizing current injections (−100 to 100 pA, 10 pA steps) illustrated that Chx10 neurons could be silenced with a few tens of pA and their firing rate began to saturate above 60 pA (*Figure 3E,F*; max rate with 100 pA: 48 ± 3.6 spikes/sec, n = 26 cells). The relatively steep slope of the FI curve between 0 and 50 pA of injected current (7.6 ± 0.8 spikes/pA) suggests that even small synaptic inputs may be sufficient to drive action potential firing in Chx10 cells. Electrically evoked excitatory and inhibitory synaptic currents (eIPSCs and eIPSCs) in Chx10 neurons could

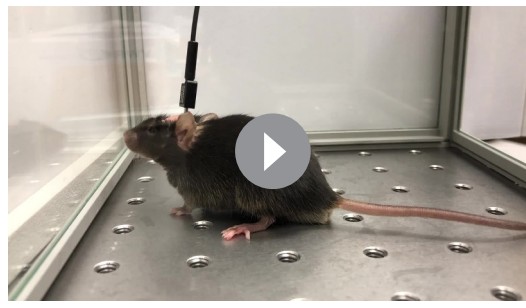

**Video 1.** Optogenetic stimulation of Chx10 neurons in vlPAG elicits freezing. Responses evoked by optogenetic activation of Chx10 neurons in the vlPAG (10 ms, 50 Hz, 5 s) is indicated by 'Light On.'.
https://elifesciences.org/articles/54302#video1

**Table 1.** Intrinsic properties of Chx10+ vlPAG neurons.

| Parameter | All Cells (n=26) | Males (n=15) | Females (n=11) | p value |
|---|---|---|---|---|
| **Intrinsic Properties** | | | | |
| Membrane Resistance (MΩ) | 584.1 ± 47.3 | 648.6 ± 62.2 | 496.0 ± 66.9 | 0.1 |
| Capacitance (pF) | 24.2 ± 1.3 | 23.4 ± 1.5 | 25.2 ± 2.5 | 0.6 |
| Interspike $V_m$ (mV)* | -48.7 ± 1.0 | -50.6 ± 1.0 | -45.9 ± 1.6 | **0.02** |
| **Action Potential Properties** | | | | |
| Spontaneous Rate (spikes/s) | 5.8 ± 1.2 | 4.5 ± 1.4 | 7.6 ± 1.9 | 0.2 |
| Maximum dV/dt (V/s) | 155.2 ± 11.4 | 153.9 ± 13.5 | 157.0 ± 20.5 | 0.9 |
| Estimated Peak $I_{Na}$ (nA) | 4.0 ± 0.5 | 3.7 ± 0.5 | 4.3 ± 0.9 | 0.6 |
| Minimum dV/dt (V/s) | -63.9 ± 5.0 | -55.1 ± 4.0 | -76.0 ± 9.5 | 0.06 |
| Estimated Peak $I_K$ (nA) | 1.6 ± 0.2 | 1.3 ± 0.1 | -2.1 ± 0.4 | 0.06 |
| Halfwidth (ms) | 1.0 ± 0.1 | 1.1 ± 0.07 | 0.9 ± 0.1 | 0.06 |
| Amplitude (mV) | 55.9 ± 2.2 | 57.3 ± 2.7 | 54.1 ± 3.5 | 0.5 |
| Threshold (mV) | -31.6 ± 0.8 | -32.5 ± 1.0 | -30.4 ± 1.4 | 0.2 |
| Rheobase at -70 mV (pA) | 9.8 ± 0.8 | 9.6 ± 1.0 | 10.0 ± 1.4 | 0.8 |

* indicates significant difference between males and females.

nevertheless be quite large. eEPSCs at −70 mV had an amplitude of −250.8 ± 7 00.1 pA and eIPSCs at 0 mV had an amplitude of 322.2 ± 90.3 pA (*Figure 3G*; n = 12 cells [12 M, 0 F]). Comparing the peak excitatory and inhibitory conductances in each cell indicated that the strength of inhibition and excitation were comparable, with a slight bias toward inhibition (*Figure 3H*; E *vs.* I: 17.6 ± 4.9 nS *vs.* 22.6 ± 6.3 nS, p=0.08, paired t-test), giving an E/I ratio of 0.78. eEPSCs decayed with a single exponential time constant, τ, of 3.5 ± 0.3 ms (*Figure 3I*) whereas eIPSCs were nearly twice as long (6.7 ± 0.6 ms; *Figure 3I*). The IPSCs may in part reflect responses to local inhibitory interneurons that tonically suppress Chx10 firing activity, and whose suppression by input from the amygdala drives freezing (*Tovote et al., 2016*). Since these synaptic responses were evoked electrically, however, their source is unknown. Since the primary interest of the present study was whether these cells might receive cerebellar signals, we next examined the properties of cells in the mCbN that were probable sources for such input.

## Intrinsic and synaptic properties of mCbN neurons

First, we recorded the properties of large, likely projection neurons in the mCbN in acute slices (*Table 2*). As in the interpositus nucleus (iCbN; *Mercer et al., 2016*; *Person and Raman, 2012*; *Raman et al., 2000*), mCbN neurons fired spontaneously at high rates (*Figure 4A*; 122.8 ± 6.6 spikes/s, n = 28 cells [15 M, 13 F]), consistent with previous reports of mCbN projection neurons in vitro (*Bagnall et al., 2009*) as well as the high basal activity of mCbN cells in vivo (*Büttner et al., 1991*; *Miller et al., 2008*; *Özcan et al., 2020*). Interestingly, spontaneous rates were higher in males (137.4 ± 8.8 spikes/s, n = 15 cells) than females (105.9 ± 9.0 spikes/s, n = 13 cells, p=0.01, unpaired t-test, *Figure 4A,B*), a difference that is in the opposite direction from the sex difference in the iCbN (*Mercer et al., 2016*). In fact, comparing the firing rates between the two nuclei showed a nearly twofold difference in males (iCbN: 72.2 ± 10.0 spikes/s, n = 15 cells, p=0.00004, unpaired t-test) but no difference for females (iCbN: 97.7 ± 9.2 spikes/s, n = 16 cells, p=0.51, unpaired t-test; iCbN data from *Mercer et al., 2016*, *Figure 4—figure supplement 1*). The high propensity for firing in the mCbN in both sexes suggests that the cerebellum likely exerts a tonic control over downstream circuitry.

Half-width analysis indicated that spontaneous action potentials were brief (*Figure 4C*; 0.26 ± 0.02 ms) and phase-plane plots estimated a threshold of −42.3 ± 0.7 mV (*Figure 4D*). Firing rates changed linearly with current injections from −300 to +300 pA (500 ms steps), with a slope of 0.36 ± 0.005 spikes/pA (*Figure 4E,F*; n = 22 cells [12 M, 10 F]). The maximum firing rate with a peak injection of +300 pA was 227.9 ± 10.9 spikes/s (n = 22 cells), with little evidence of saturation,

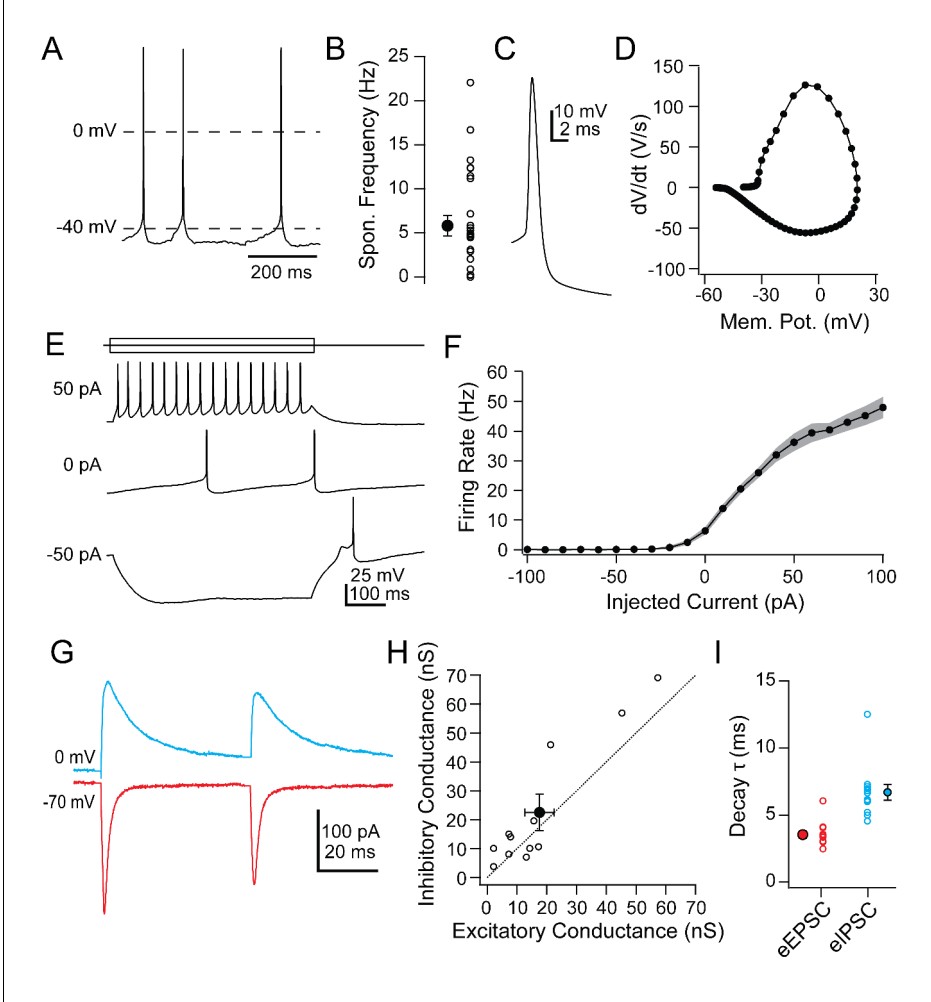

**Figure 3.** Intrinsic and synaptic properties of Chx10 neurons in PAG slices. (A) Spontaneous action potentials from a Chx10 neuron. (B) Population data for spontaneous firing rates of Chx10 neurons. (C) Single spontaneous action potential of a Chx10 neuron. (D) Phase-plane plot of action potential in C. (E) Action potentials evoked by 500 ms current injections of −50, 0 and 50 pA in a Chx10 neuron. (F) Mean FI curve for all neurons. *Solid symbols*, mean, *grey shading*, SEM. (G) Evoked EPSCs (*red*) and IPSCs (*blue*) from a single Chx10 neuron. (H) Peak inhibitory conductance *vs.* peak excitatory conductance. *Open symbols,* individual cells, *solid symbols,* mean ± SEM. *Dotted line*, unity. (I) Decay time constants, τ, of EPSCs and IPSCs. *Open symbols,* individual cells, *solid symbols,* mean ± SEM.

The online version of this article includes the following figure supplement(s) for figure 3:

**Figure supplement 1.** Comparison of Chx10 and GAD2 intrinsic properties.

suggesting that mCbN neuron firing rates can be elevated as well as suppressed over a wide dynamic range.

In fact, large IPSCs were evoked in mCbN neurons by stimulating Purkinje cells either electrically (n = 7 cells [3 M, 4 F]) or with light in L7-ChR2 mice (n = 7 cells [0 M, 7 F]); since both procedures elicited comparable responses, the data were pooled, (*Figure 4G,H*). IPSCs at 0 mV were 3.0 ± 2.7 nA, reflecting a maximal conductance of 42.5 ± 10.2 nS, and had a τ of 4.2 ± 0.3 ms (*Figure 4I*), a value that is significantly longer than in the iCbN (*Figure 4—figure supplement 1*; iCbN: 2.14 ± 0.1 ms, n = 34 cells [22 M, 12 F], p<0.00001, unpaired t-test; iCbN data from *Mercer et al. (2016)*. The relatively slower kinetics may make mCbN cells more readily suppressed by afferent Purkinje cells, since the efficacy of inhibition is highly dependent on IPSC time course (*Person and Raman, 2012*; *Najac and Raman, 2015*; *Wu and Raman, 2017*; *Brown and Raman, 2018*).

**Table 2.** Intrinsic properties of medial cerebellar nucleus neurons.

| Parameter | All Cells (n=28) | Males (n=15) | Females (n=13) | p value |
|---|---|---|---|---|
| **Intrinsic Properties** | | | | |
| Membrane Resistance (MΩ) | 77.5 ± 12.8 | 62.2 ± 8.1 | 95.5 ± 17.0 | 0.08 |
| Capacitance (pF) | 68.6 ± 5.5 | 79.7 ± 12.6 | 55.2 ± 5.8 | 0.09 |
| **Action Potential Properties** | | | | |
| Spontaneous Rate (spike/s)* | 122.8 ± 6.6 | 137.4 ± 8.8 | 105.9 ± 9.0 | **0.01** |
| Maximum dV/dt (V/s) | 294.3 ± 19.0 | 270.6 ± 16.6 | 321.8 ± 34.6 | 0.2 |
| Estimated Peak $I_{Na}$ (nA) | 19.4 ± 2.0 | 20.8 ± 2.9 | 17.9 ± 2.8 | 0.5 |
| Minimum dV/dt (V/s) | -270.7 ± 20.1 | -242.1 ± 19.4 | -303.4 ± 38.4 | 0.2 |
| Estimated Peak $I_K$ (nA) | 17.8 ± 1.8 | 18.4 ± 2.4 | 17.2 ± 3.0 | 0.7 |
| Halfwidth (ms) | 0.27 ± 0.02 | 0.27 ± 0.2 | 0.27 ± 0.4 | 0.9 |
| Ampliutde (mV) | 53.2 ± 1.5 | 51.7 ± 1.7 | 54.9 ± 2.8 | 0.3 |
| Threshold (mV) | -42.3 ± 0.7 | -41.7 ± 1.0 | -43.0 ± 1.1 | 0.4 |

* indicates significant difference between males and females.

## Effects of mCbN input to the vlPAG

Given that Purkinje cells inhibit mCbN cells, along with the observation that lesions of the cerebellar vermis reduce innate freezing (*Supple et al., 1988*; *Koutsikou et al., 2014*), the simplest prediction is that mCbN input might suppress the activity of Chx10 cells. Such an effect might be achieved by direct inhibition of Chx10 cells, since a subset of mCbN cells have been reported to be glycinergic (*Bagnall et al., 2009*) or by excitation of other neurons that lead to a decrease in Chx10 cell activity. Since cerebellar output is not consistently the inverse of Purkinje cell activity, however (*Armstrong and Edgley, 1984a*; *Armstrong and Edgley, 1984b*; *Brown and Raman, 2018*), the converse may instead be true. Therefore, to test whether mCbN neurons form synapses directly onto Chx10 neurons, we expressed ChR2 in mCbN afferents through viral injection into the mCbN and recorded either in wild-type mice from vlPAG cells whose molecular phenotype was unidentified, or from identified Chx10 or GAD2+ vlPAG cells in mice with cell-specific labels. Light stimulation of ChR2-labeled mCbN axons in the vlPAG elicited EPSCs at −70 mV in only a subset of unidentified (11%, n = 4 of 38 cells), Chx10 (20%, n = 5 of 25 cells), and GAD2+ (21%, n = 3 of 14 cells) cells. In cells in which an EPSC was evoked, the direct mCbN-evoked synaptic currents had similar amplitudes across all three categories (*Figure 5A,B*; unidentified: −40.6 ± 15.4 pA, n = 4 cells; Chx10: −33.9 ± 9.7 pA, n = 5 cells; GAD2+: 39.6 ± 16.7 pA, n = 3 cells). Regardless of whether or not the cell received a direct EPSC, mCbN stimulation never evoked detectable IPSCs (n = 42 of 42 cells). Thus, although some mCbN projection neurons are glycinergic (*Bagnall et al., 2009*), the projection to the vlPAG is glutamatergic. Since Chx10 neurons appear to act essentially as premotor neurons, cerebellar excitation of these cells may have the potential to facilitate (or even elicit) freezing under some conditions. Despite the comparable excitation of GAD2 cells, the lack of mCbN-dependent IPSCs in any recorded cell type suggests that the mCbN projection does not effectively recruit polysynaptic local inhibition.

Nevertheless, the connectivity ratio of about 20% (for Chx10 and GAD2 cells) seemed moderate, particularly given the anatomical evidence for a relatively strong projection. We therefore considered the possibility that the cerebellum may convey information to the vlPAG through a modulatory signal, thereby altering the integration of inputs in the vlPAG microcircuit. To test this possibility, we stimulated ChR2-expressing mCbN afferents at 25 Hz for near-maximal release of modulatory neurotransmitters (*Vaaga et al., 2017*) and measured electrically evoked PSCs in Chx10 neurons. Indeed, mCbN optogenetic stimulation increased the amplitude of eIPSCs in Chx10 neurons from 233.1 ± 49.9 pA to 280.7 ± 53.4 pA (*Figure 5C*, n = 12 cells [10 M, 2 F]), corresponding to a within-cell increase of 28.7 ± 07.0% (*Figure 5D*, *left*; *p=0.0016*, one-sample t-test). The increased IPSC amplitude occurred without a corresponding change in the paired-pulse ratio (PPR), which went from 1.03 ± 0.08 to 1.02 ± 0.07 (*Figure 5D*, *right*, n = 12 cells; *p=0.94*, paired t-test). Additionally,

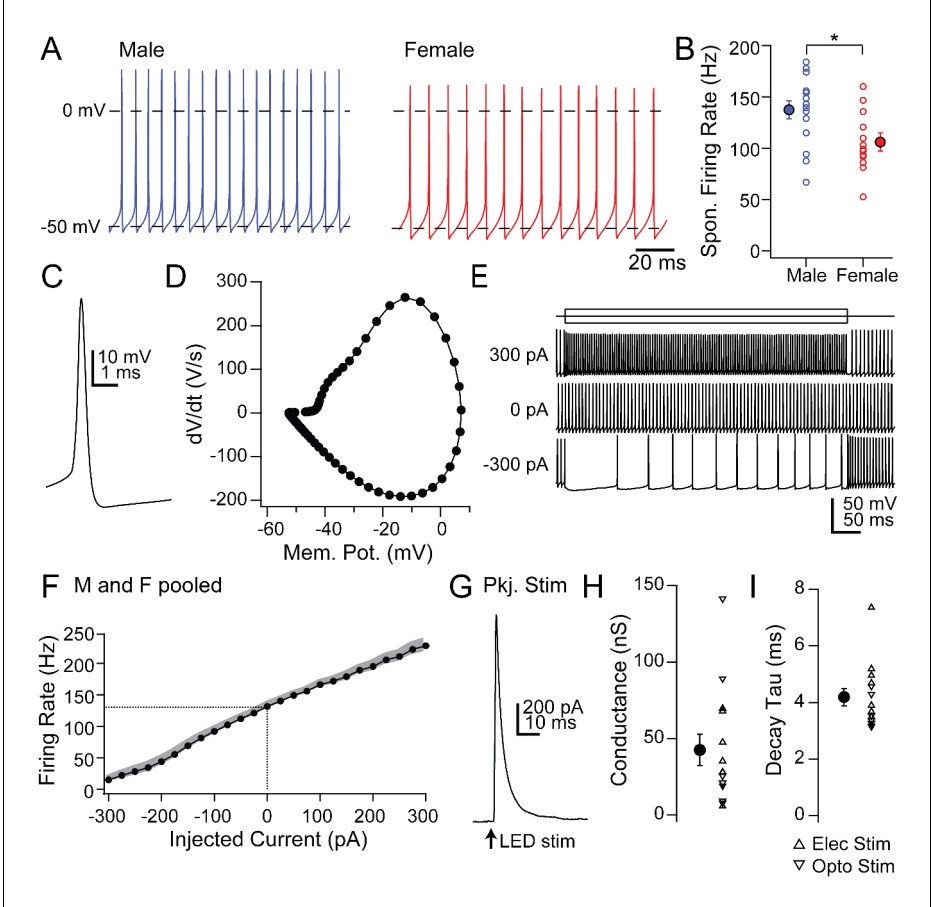

**Figure 4.** Intrinsic and synaptic properties of mCbN neurons in cerebellar slices. (**A**) Spontaneous action potentials recorded from mCbN neurons from males (*blue*) and females (*red*). (**B**) Population data of spontaneous firing rates separated by sex. *Open symbols,* individual cells, *solid symbols,* mean ± SEM. *Asterisks, p<0.05.* (**C**) Single spontaneous action potential of an mCbN neuron. (**D**) Phase-plane plot of action potential in C. (**E**) Action potentials evoked by 500 ms current injections of −300, 0 and 300 pA in an mCbN neuron. (**F**) Mean FI curve for all neurons from both male and female mice. *Solid symbols,* mean, *grey shading*, SEM. *Dotted line,* firing rate at 0 pA injected current. (**G**) IPSC in an mCbN cell evoked by optogenetic stimulation of Purkinje cells. (**H**) Population data of IPSC conductances. *Upward triangles,* electrical stimulation; *downward triangles,* optogenetic stimulation. *Open symbols,* individual cells, *solid symbols,* mean ± SEM. (**I**) Population weighted decay time constants for evoked IPSCs in mCbN neurons. *Open symbols,* individual cells, *solid symbols,* mean ± SEM.

The online version of this article includes the following figure supplement(s) for figure 4:

**Figure supplement 1.** Comparison of mCbN and iCbN intrinsic and synaptic properties in males and females.

mCbN stimulation had the converse effect on eEPSCs in Chx10 cells, decreasing them from −401.4 ± 55.2 pA to −355.6 ± 55.3 pA (*Figure 5E*, n = 12 cells [3 M, 9 F]), a reduction of 13.7 ± 4.3% (*Figure 5F*, *left*; *p=0.0075*, one sample t-test), again without a change in PPR (*Figure 5F*, *right*; 0.83 ± 0.07 to 0.88 ± 0.07; p=0.15, paired t-test). Modulation of PSC amplitude by mCbN stimulation was reliable, leading to a change of >10% in the majority of Chx10 cells (11/12 for IPSCs and 9/12 for EPSCs).

To quantify the relative effect of modulation by mCbN stimulation, we calculated the EI ratio as the ratio of the excitatory to the inhibitory conductance (*Eichler and Meier, 2008*; *Turrigiano and Nelson, 2004*). Before stimulation, the within-cell EI ratio in Chx10 cells was 0.78 (17.6/22.6 nS); the mean percent change in excitation and inhibition predicts that, after stimulation, the EI ratio would fall to 0.53, i.e., a 32.5% decrease. These data suggest that mCbN afferents reliably modulate the strength of both IPSCs and EPSCs to favor inhibition. Given the high basal activity of mCbN cells, it

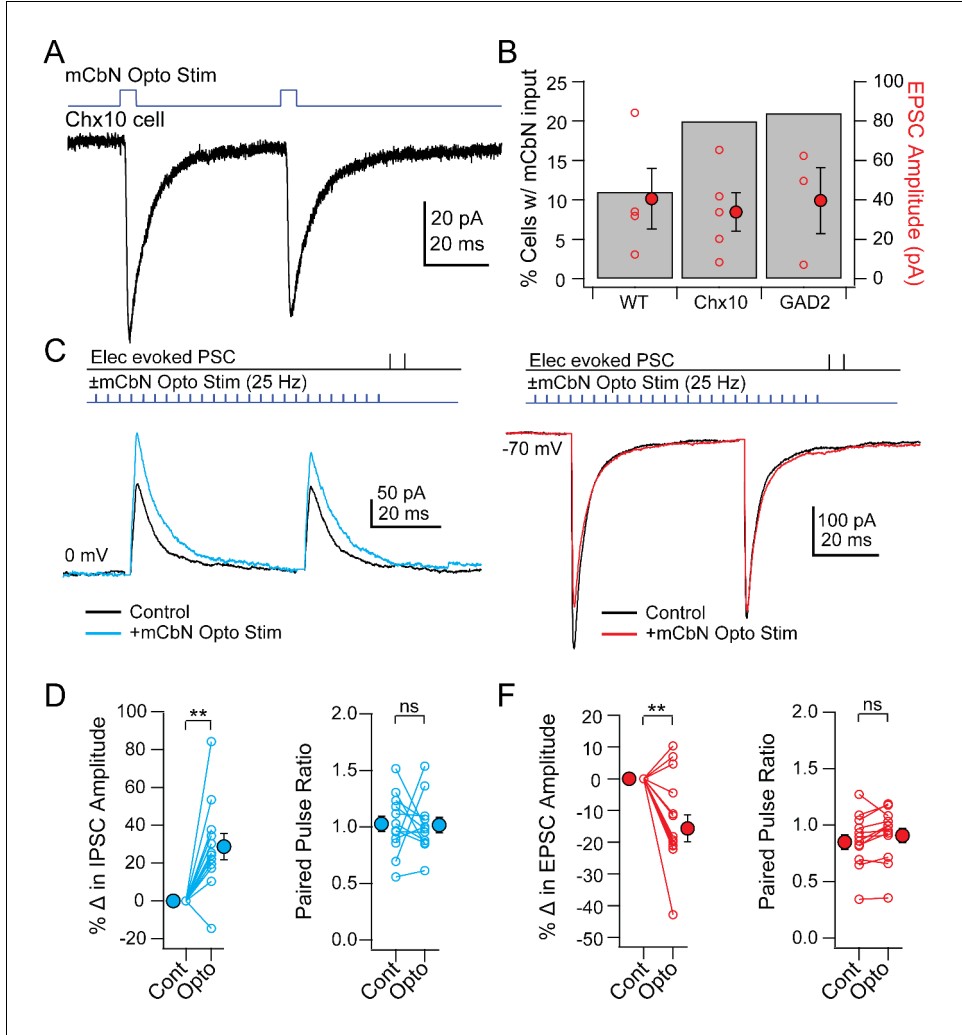

**Figure 5.** Direct and modulated PSCs in Chx10 cells evoked by mCbN optogenetic stimulation. (**A**) EPSCs evoked in a Chx10 cell by optogenetic stimulation of mCbN afferents. (**B**) Percentage of tested cells in which responses could be elicited (grey bars) and population data for EPSC amplitudes (red circles), for unlabeled vlPAG cells (WT), Chx10 neurons, and GAD2 neurons. Open symbols, individual cells, *solid symbols,* mean ± SEM. (**C**) *Top*, protocol for stimulating electrically (upper line) and optogenetically (*lower line*). *Bottom*, IPSCs without (black) and with (color) mCbN stimulation. (**D**) Population data for percent change in IPSC (left) and PPR (right) without (Cont) and with mCbN (Opto) stimulation. *Open symbols,* individual cells, *solid symbols,* mean ± SEM. *Asterisks, p<0.01,* n.s., non-significant. (**E, F**) As in C, D for EPSCs.

seems possible that the cerebellum exerts a tonic control of synaptic strength in Chx10 cells, biasing these neurons toward a more inhibited state, which is expected to favor movement.

## Mechanism of mCbN-induced modulation of postsynaptic currents

The vlPAG includes a mixed population of dopaminergic and noradrenergic tyrosine hydroxylase (TH) positive neurons (*Figure 6A*; *Suckow et al., 2013*), raising the possibility that these neurons might mediate the mCbN-dependent modulation of synaptic input to Chx10 neurons. To test whether TH neurons are in fact targets of mCbN neurons, we recorded from labeled TH+ neurons in TH-tdT mice (*Figure 6A*) in which ChR2-eYFP had been virally introduced into mCbN neurons. The connectivity ratio for mCbN cells onto TH vlPAG neurons was higher than onto Chx10 cells, with 72.7% of TH neurons (n = 8 of 11 cells [8 M, 0 F]) responding to optogenetic stimulation of mCbN afferents with EPSCs (*Figure 6B,C*, *left*; −35.2 ± 8.7 pA, n = 8), with a PPR of 0.8 ± 0.2 (*Figure 6C*, *right*; n = 8 cells). These TH neurons fired spontaneously (17.9 ± 4.6 sp/s, n = 8 cells [8 M, 0F]), as

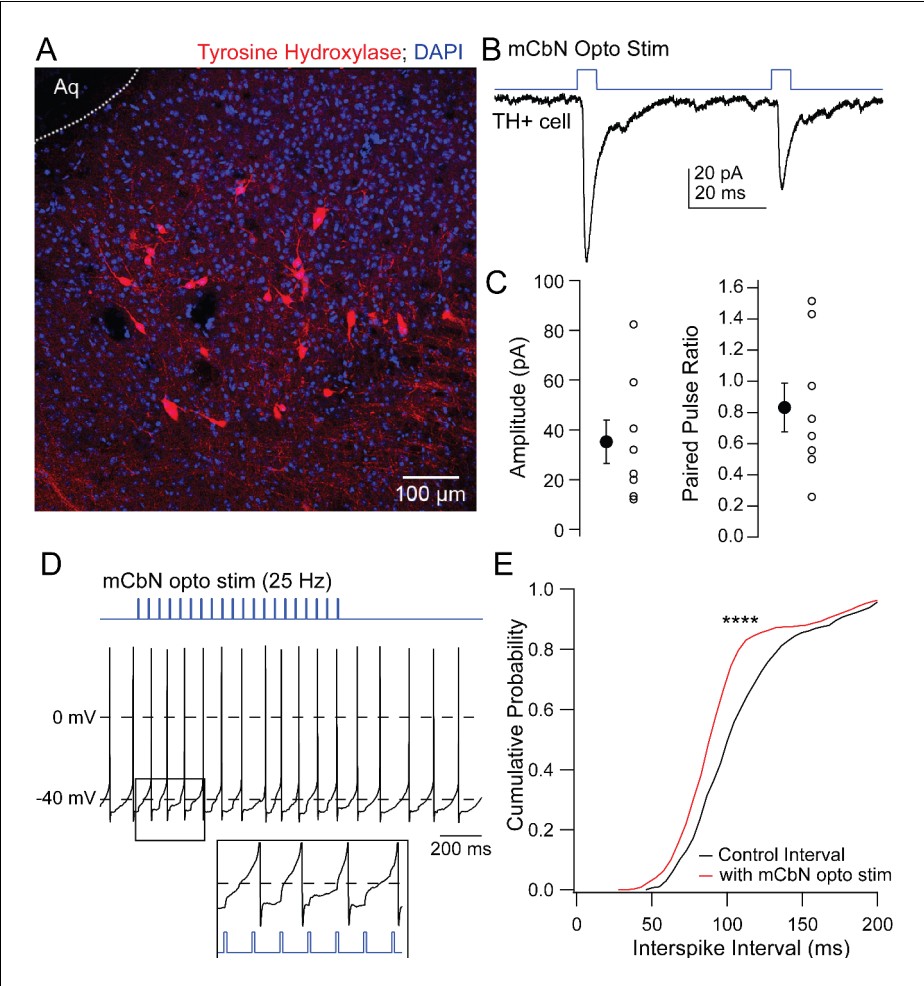

**Figure 6.** EPSCs in TH neurons evoked by mCbN activation. (**A**) Distribution of TH neurons in the vlPAG. (**B**) Optogenetically evoked EPSCs in a TH neuron. (**C**) Population data for the first EPSC (left) and PPR (right) evoked as in B. Open symbols, individual cells, solid symbols, mean ± SEM. (**E**) Stimulus protocol (*top*) and action potentials in a TH neuron (*bottom*). *Inset,* magnification of stimulus-evoked EPSPs. (**F**) Cumulative probability distribution of interspike intervals before (black) and during (red) mCbN stimulation. Asterisks, p<0.0001.

previously reported (*Dougalis et al., 2017*). The magnitude of the mCbN-dependent synaptic input evoked by 30 light pulses at 25 Hz was indeed sufficient to accelerate spike onset (*Figure 6D*), leading to a leftward shift of the cumulative interspike interval distribution (*Figure 6E*; control ISI: 112.7 ± 1.5 ms; +mCbN stimulated ISI: 100.0 ± 1.3 ms, n = 4 cells [4 M, 0 F]; p<0.0001, Kolmogorov-Smirnov test). The leftward shift in the interspike interval distribution corresponded with a modest increase in firing rate (control: 9.9 ± 0.1 spikes/sec; +mCbN stimulation: 11.2 ± 0.1 spikes/sec, n = 4 cells [4M, 0F]).

If mCbN-induced modulation of Chx10 neurons is mediated through local TH neurons, then directly activating TH neurons should mimic the synaptic effects of elevating mCbN activity. To test this prediction, we began by stimulating ChR2-expressing TH neurons ('TH-ChR2') while recording PSCs from unlabeled vlPAG neurons. Although TH neurons might be either noradrenergic or dopaminergic, pilot studies showed that bath-applied isoproterenol, a β-adrenergic agonist, did not mimic the effect of mCbN stimulation, and instead reduced IPSC amplitudes. Therefore, we isolated the effect of dopamine by making recordings in the presence of α- and β-adrenergic receptor antagonists (5 µM prazosin, 30 µM sotalol). When the stimulus trains previously applied to mCbN cells were applied to TH-ChR2 cells, PSCs in unlabeled neurons were modulated similarly: eIPSCs

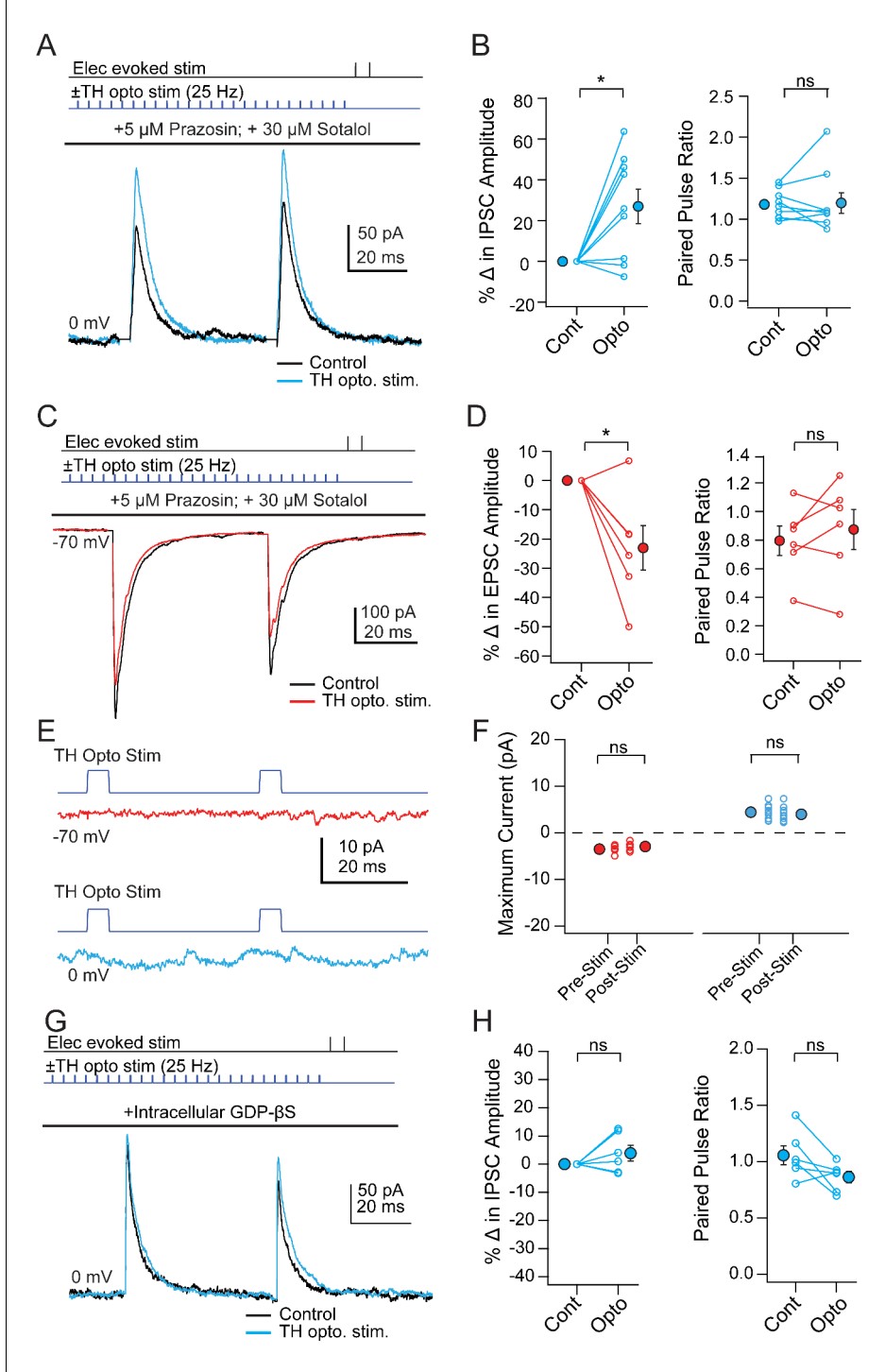

**Figure 7.** Modulation of PSCs in unlabeled vlPAG cells evoked by TH neuron stimulation. (**A**) *Top*, protocol for stimulating electrically (upper line) and optogenetically (lower line). Bottom, IPSCs without (black) and with (color) mCbN stimulation. (**B**) Population data for percent change in IPSC (left) and PPR (right) without (Cont) and with mCbN (Opto) stimulation. *Open symbols*, individual cells, *solid symbols*, mean ± SEM. *Asterisks, p<0.05,* n.s., non-significant. (**C, D**) As in A, B for EPSCs. (**E**) Optogenetic stimulus to TH-ChR2 neurons and lack of direct synaptic responses at −70 mV (*top*) and 0 mV (*bottom*). (**F**) Population data for the absence of synaptic responses at −70 mV (*left*) and 0 mV (*right*) evoked by optogenetic TH-ChR2 neuron stimulation. (**G, H**) As in A, B for IPSCs with intracellular GDP-βS.

increased in amplitude from 139.5 ± 12.3 pA to 175.5 ± 17.3 pA; (*Figure 7A*, n = 9 cells [2 M, 7 F]), giving a within-cell increase of 27.0 ± 8.5% (*Figure 7B*, *left; p=0.013*, one sample t-test) with no change in PPR (*Figure 7B*, *right*; 1.8 ± 0.06 *vs.* 2.0 ± 0.1, n = 9, p=0.84, paired t-test). Likewise, eEPSCs decreased from −423.9 ± 85.9 pA to −340.9 ± 95.8 pA (*Figure 7C*, n = 6 cells [4 M, 2 F]), corresponding to a within-cell drop of 23.1 ± 7.7% (*Figure 7D*, *left; p=0.03,* one sample t-test) with no change in PPR (*Figure 7D*, *right*; 0.8 ± 0.1 *vs.* 0.9 ± 0.1, p=0.4, paired t-test). Stimulation of TH-ChR2 neurons, however, did not elicit direct EPSCs (*Figure 7E*, *top*, **F**; peak pre-stimulus inward current: −3.5 ± 0.3 pA, peak post-stimulus inward current: −3.0 ± 0.4 pA p=0.3, n = 6 cells) or IPSCs (*Figure 7E*, *bottom,* **F**; peak pre-stimulus outward current: 4.4 ± 0.5 pA, peak post-stimulus outward current, 4.0 ± 0.6 pA, p=*0.4*, n = 9 cells) in unlabeled cells. Local co-release of glutamate or GABA from TH neurons therefore seems unlikely. As this population of dopaminergic neurons has been shown to co-release glutamate in the central amygdala (*Groessl et al., 2018*), the results are consistent with spatial segregation of co-transmission (*Vaaga et al., 2014*).

Because the PPR of PSCs was stable throughout modulation by both mCbN neurons and TH cells, we tested whether the site of modulatory action was postsynaptic by repeating the experiments with GDP-βS (500 μM) included in the pipette to occlude G-protein signaling (*Eckstein et al., 1979*). This manipulation blocked the effects of TH cell stimulation on eIPSCs, such that responses were 171.0 ± 26.5 pA before and 176.1 ± 26.0 pA after stimulation (*Figure 7G*, n = 6 cells [4 M, 2 F]). The within-cell change was thus reduced to 3.8 ± 2.8% (*Figure 7H*, *left; p=0.24,* one sample-t-test), with PPR remaining near 1 (*Figure 7H*, *right*; 1.1 ± 0.09 *vs.* 0.9 ± 0.1; p=0.07, paired t-test). The observation that GDP-βS was sufficient to prevent the modulation of IPSCs provides further evidence that the increase in IPSC amplitudes results from an altered postsynaptic response to GABA, rather than from polysynaptic recruitment of additional inhibitory afferents to the recorded neurons. Taken together, these data suggest that local TH interneurons have the capacity to mediate the mCbN-dependent modulation of synaptic inputs via a postsynaptic action of dopamine.

To investigate the receptor subtypes responsible for the changes in PSCs specifically in Chx10 cells, we tested whether modulation could be mimicked by selective activation of either $D_1$ or $D_2$ receptors. Much like stimulation of TH-ChR2 cells, bath application of the $D_2$ receptor agonist quinpirole (25 μM) increased IPSC amplitudes in Chx10 cells from 140.6 ± 13.8 pA to 217.8 ± 29.5 pA (*Figure 8A*; n = 5 cells [2 M, 3 F]), a within-cell increase of 53.3 ± 10.4% (*Figure 8B*; p**=0.007**, one sample t-test) and decreased eEPSCs from −262.1 ± 99.8 pA to −191.9 ± 87.7 pA (*Figure 8C*; n = 5 cells [1 M, 4 F]), a within-cell drop of 34.3 ± 11.3% (*Figure 8D*; p=0.04, one sample t-test). In contrast, selectively activating $D_1$ receptors, by bath application of 10 μM dopamine with the $D_2$ antagonist sulpiride (1 μM), did not significantly change PSC amplitudes. Activation of $D_1$ receptors resulted in a within-cell change of 11.2 ± 17.7% in eIPSCs (*Figure 8A,B*, n = 5 cells [4 M, 1 F], p=*0.8*, one sample t-test) and a −0.9 ± 8.7% change in eEPSCs (*Figure 8C,D*; n = 5 cells [2 M, 3 F], p=*0.6*, one sample t-test). These data demonstrate that the dopamine-dependent modulation of PSCs in Chx10 neurons is likely mediated by postsynaptic $D_2$ receptors.

Finally, we tested whether $D_2$ receptors also mediate the effects of mCbN activation on Chx10 synaptic responses. To do so, we stimulated ChR2-expressing mCbN afferents in the absence and presence of the $D_2$ receptor antagonist sulpiride (*Figure 9A,C*). Since neurons could not withstand the repeated depolarizations and repolarizations necessary to record EPSCs and IPSCs in a single cell, we recorded only EPSCs or IPSCs in each cell and compared the extent of modulation in the presence of sulpiride to that seen in $D_2$-available control conditions. This control dataset included 24 cells recorded in *Figure 5*, as well as 3 cells recorded with $D_1$ receptors blocked. Since the prediction was that no modulation would be seen, we took advantage of the fact that, in previous experiments, PSC modulation of different cells could be obtained sequentially in the same slice. Therefore, as a positive control, we first confirmed in each slice that mCbN stimulation indeed elicited a change in PSC strength before recording subsequent cells within the same slice in the presence of $D_1$ or $D_2$ receptor antagonists, adding 8 cells in eight slices to the control dataset, for a total of 35 cells (18 for IPSCs and 17 for EPSCs).

As reported above, in ACSF-treated control cells mCbN stimulation increased eIPSCs from 161.9 ± 49.6 pA to 208.6 ± 60.8 pA (*Figure 9B,D*), a 30.4 ± 9.1% within-cell change (n = 4 cells [4 M, 0 F], p=0.04, one sample t-test) and decreased eEPSCs from −222.5 ± 38.5 pA to −183.7 ± 40.5 pA, a −19.2% within-cell change (n = 4 cells [3 M, 1 F], p=0.02, one sample t-test). Application of the $D_1$ antagonist SCH-23390 (1 μM) in 3 cells gave values that fell within the control distribution

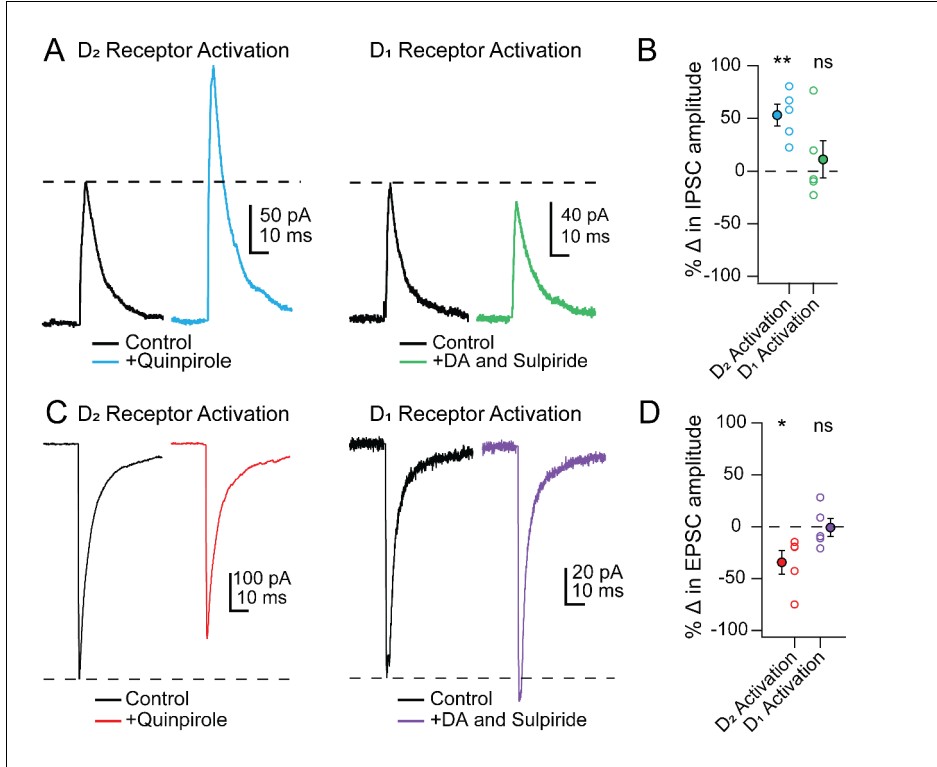

**Figure 8.** PSC modulation in Chx10 cells by $D_2$ but not $D_1$ receptor activation. (**A**) Left, Electrically evoked IPSCs before (black) and during (color) bath application of the $D_2$-receptor agonist quinpirole (25 μM). *Right*, As at left, but with $D_1$ receptors activated by dopamine while $D_2$ receptors were blocked by sulpiride (1 μM). (**B**) Population data for percent change in IPSC with either $D_2$ or $D_1$ receptor activation. *Open symbols,* individual cells, *solid symbols,* mean ± SEM. *Asterisks, p<0.01,* n.s., non-significant. (**C**, **D**) As in A, B but for EPSCs.

(*Figure 9B,D*). Bath application of sulpiride (1 μM), however, blocked both effects, such that in none of the 10 cells recorded in sulpiride did the extent of change reach the mean value for the modulated case. The eIPSCs of 246.8 ± 65.4 pA before stimulation remained at 246.4 ± 53.8 pA, a 3.1 ± 5.3% within-cell change (*Figure 9A,B*; n = 5 cells [5 M, 0 F], p=0.6, one sample t-test, % change *vs.* ACSF, p=0.03, unpaired t-test) and eEPSCs of −120.0 ± 14.7 pA before stimulation stayed at −122.1 ± 17.5 pA, a 1.0 ± 3.2% within-cell change (*Figure 9C,D*; n = 5 cells [3 M, 2 F], p=0.8, one sample t-test; % change *vs.* ACSF, p=0.005, unpaired t-test). Together, these results suggest that that the mCbN-induced modulation of IPSC and EPSC strength onto Chx10 cells is mediated through activation of local dopaminergic interneurons within the vlPAG, via postsynaptically expressed $D_2$ receptors.

## Discussion

Here, we demonstrate that the cerebellum directly regulates midbrain regions that drive innate freezing in mice, via glutamatergic projections from the mCbN that influence multiple classes of neurons in the vlPAG (schematized in *Figure 10*). We find that mCbN afferents can evoke EPSCs in glutamatergic Chx10-expressing neurons, whose activity is shown to be sufficient to generate freezing in the absence of threat in freely moving mice, as well as in GAD2 neurons, which are expected to provide local inhibition. The connection probability onto both these cell types is ~20%, suggestive of a bidirectional, though relatively sparse, participation of the mCbN in fast synaptic transmission in local circuits that produce freezing. The denser cerebellar input appears to be modulatory, as the mCbN contacts TH-expressing neurons in the vlPAG with a ~ 70% connection probability. Repetitive stimulation of mCbN afferents increases IPSCs and decreases EPSCs in Chx10 cells through activation of dopamine $D_2$ receptors, which can be mimicked by direct stimulation of vlPAG TH neurons.

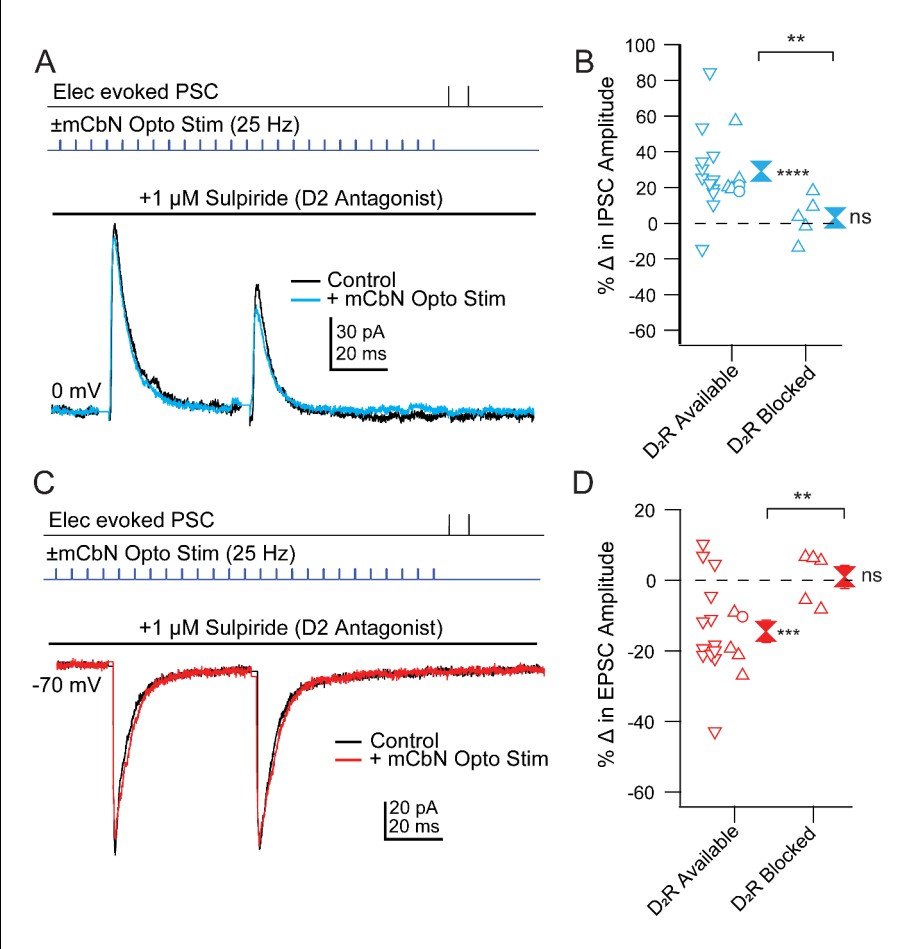

**Figure 9.** Blockade of mCbN-dependent modulation of Chx10 neuron PSCs by $D_2$ receptor antagonists. (**A**) *Top*, protocol for stimulating electrically (upper line) and optogenetically (lower line). Bottom, IPSCs without (black) and with (color) mCbN stimulation in the presence of sulpiride. (**B**) Population data for percent change in IPSC with $D_2$ receptors available or blocked. *Open symbols*, individual cells. Control data with $D_2$ receptors available include cells with no antagonists from *Figure 5D* (*downward triangles*), cells with $D_1$ antagonists (*open circles*), and cells from slices in which $D_2$ receptors were subsequently blocked (*upward triangles*). *Solid symbols*, mean of *up* and *down triangles* ± SEM. *Two asterisks, $p<0.01$ (unpaired t-test with and without $D_2$ receptors blocked), three asterisks, $p<0.001$ (one sample t-test), n.s., non-significant (one sample t-test). (**C, D**) As in A, B for EPSCs.

Recordings from mCbN neurons demonstrate that they have a wide dynamic range with high spontaneous firing rates that differ between sexes. Cerebellar input thus appears likely to regulate dopaminergic tone in the vlPAG, shifting the relative strength of synaptic inputs onto Chx10 neurons to favor inhibition. Since we find that Chx10 neurons are likely to generate freezing by directly exciting the Mc, the simplest interpretation is that mCbN activity stimulates dopamine release that biases the local circuit toward inhibition of Chx10 cells, thereby being permissive of movement. Such a scenario would provide a mechanistic explanation for behavioral studies implicating activation of the cerebellar vermis, hence inhibition of the mCbN, as facilitating innate freezing.

*Identification of Chx10 neurons in the vlPAG as drivers of freezing.* The periaqueductal gray regulates a range of defensive behaviors, as activation of distinct rostro-caudal columns within the PAG elicits distinct responses such as flight (lPAG) or freezing (vlPAG) (*Carrive, 1993*; *Bandler and Shipley, 1994*; *Behbehani, 1995*; *Bandler et al., 2000*; *Keay and Bandler, 2001*). In vivo, optogenetic activation of glutamatergic vlPAG neurons that project to the Mc elicits freezing without analgesia in mice, whereas silencing of this cell population occludes innate freezing in response to threatening stimuli (*Tovote et al., 2016*). Additionally, when threat probability is not all or none, firing rates in subsets of glutamatergic vlPAG neurons in rats correlate with the degree of danger, as do

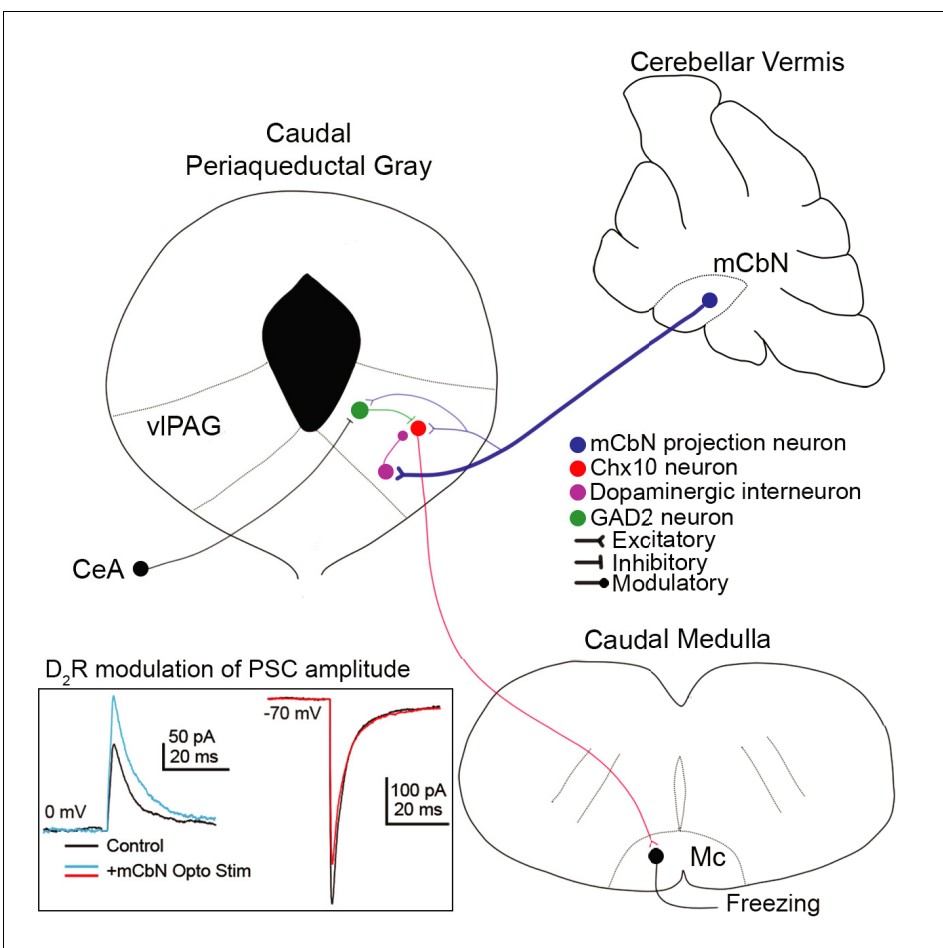

**Figure 10.** Cerebellar influence on the vlPAG freezing-related microcircuit. (**A**) mCbN afferents (blue) excite Chx10 (red), GAD2 (green) and TH (purple) neurons in the vlPAG, with the strongest functional connection occurring between mCbN neurons and TH neurons. Chx10 neurons excite the magnocellular reticular nucleus (Mc) of the caudal medulla, which drives freezing output. Within the vlPAG, TH neurons modulate PSCs on Chx10 neurons via $D_2$ receptors (*inset, traces overlaid from* ***Figure 5***).

behavioral responses (***Wright and McDannald, 2019***). The heterogeneous activity of glutamatergic vlPAG neurons, with distinct classes of response patterns evident, suggests the existence of multiple classes of glutamatergic vlPAG cells, which likely contribute differentially to assess threat probability and generate freezing (***Wright and McDannald, 2019***).

Here, we have identified a particular subset of these glutamatergic neurons that express the transcription factor Chx10 and are well-suited to generate freezing responses. Chx10 neurons project to the Mc, which is directly upstream of spinal cord motor neurons (***Esposito et al., 2014***; ***Tovote et al., 2016***). Likewise, optogenetic stimulation of Chx10 neurons in the vlPAG reliably elicits robust freezing (***Leiras et al., 2017***, SfN abstract). Consistent with a role in directly driving freezing, we find that the duration of immobility and the duration of light stimulation of Chx10 cells are precisely correlated, and freezing does not habituate over repeated trials. These data suggest that Chx10 neurons directly evoke freezing motor programs. Because the light stimulation was selected to produce high firing rates, however, more natural stimulation that modulates the activity of these cells throughout their dynamic range might have the capacity to generate more graded behavioral responses.

## Cerebellar connections to brain regions regulating complex behaviors

The cerebellum has long been recognized as being involved in motor control, as disruptions of its signals give rise to disorders of movement, and experimentally elevating firing rates of CbN neurons

can trigger muscle contraction (*Noda and Fujikado, 1987*; *Hesslow, 1994a*; *Witter et al., 2013*; *Heiney et al., 2014*; *Proville et al., 2014*; *Lee et al., 2015*). More generally, cerebellar activity contributes to prediction on a sub-second timescale (*Ivry et al., 2002*; *Mauk and Buonomano, 2004*; *Molinari et al., 2008*). It is also becoming clear that the cerebellum contributes to behaviors that carry a valence of reward or aversion (*Kostadinov et al., 2019*; *Heffley and Hull, 2019*; *Carta et al., 2019*). In doing so, it communicates widely with many regions of the brain. The CbN projects directly to brainstem centers to alter, generate, or augment reflexes, such as those associated with the vestibulo-ocular reflex, eyelid conditioning, and whisking (*Lisberger, 1988*; *Medina et al., 2000*; *Brown and Raman, 2018*); the cerebellum also forms multisynaptic loops with the cerebral cortex and hippocampus to modulate complex behaviors, including sensory discrimination and navigation (*Popa et al., 2013*; *Proville et al., 2014*; *Rochefort et al., 2011*; *Babayan et al., 2017*).

In conjunction with published studies, the present data provide evidence that the cerebellum also forms loops with the circuitry that mediates defensive behaviors. Indeed, a recent report provides chemogenetic evidence that the mCbN projection to the vlPAG bidirectionally modulates expression of fear memory in awake behaving mice (*Frontera et al., 2020*, BioRxiv). Previous studies have shown that the cerebellar cortex receives and responds to signals from the vlPAG via both mossy fiber and climbing fiber inputs (*Watson et al., 2013*; *Koutsikou et al., 2014*; *Dietrichs, 1983*; *Apps and Strata, 2015*). The vlPAG functionally connects to vermal regions of the cerebellum, and local cerebellar lesions of these areas disrupt learned fear responses (*Koutsikou et al., 2014*). Moreover, synaptic plasticity in the cerebellar cortex is evident during fear conditioning training (*Sacchetti et al., 2002*; *Sacchetti et al., 2004*; *Farley et al., 2016*), consistent with the well-studied roles of Purkinje cells and their targets in motor learning (*Medina et al., 2002*). Interestingly, however, lesions of vermal Purkinje cells (*Supple et al., 1988*; *Koutsikou et al., 2014*), which are expected to increase activity in the mCbN, also interfere with innate freezing, suggesting that cerebellar output facilitates movement. This observation is reminiscent of evidence that directly suppressing cerebellar output from other cerebellar nuclei can halt or slow movement even in non-associative complex or reflexive motor behaviors (*Hesslow, 1994b*; *Goodkin and Thach, 2003*; *Sarnaik and Raman, 2018*; *Brown and Raman, 2018*).

*Effects of mCbN input on neurons of the vlPAG.* Here, we find that the mCbN projects directly to the vlPAG, where it exerts two distinct effects on local microcircuits. First, mCbN afferents directly excite a subpopulation of Chx10 neurons, a connection that appears suited for directly driving or facilitating freezing. Also, however, the mCbN directly excites GAD2 interneurons; these cells may be among the GABAergic neurons that play a central role in driving conditioned fear responses, as they are inhibited by the central amygdala, thus disinhibiting the freezing-related cells that project to the Mc (*Tovote et al., 2016*; *Oka et al., 2008*). Although the finding of mCbN-mediated EPSCs in both Chx10 and GAD2 neurons makes it difficult to generalize about the cerebellar influence on freezing, the present data suggest that excitation from the mCbN, possibly from different cell types, is positioned to regulate both innate and learned freezing.

The second, functionally more robust means by which the cerebellum modulates vlPAG circuitry is through modulation of fast synaptic transmission: activation of mCbN afferents, or stimulation of local TH neurons, simultaneously amplifies IPSCs while reducing EPSCs in Chx10 neurons. Interestingly, TH neurons in the PAG are implicated in several behaviors, including anti-nociception (*Meyer et al., 2009*; *Li et al., 2016*; *Taylor et al., 2019*), sociability (*Matthews et al., 2016*), fear learning (*Groessl et al., 2018*), and arousal (*Porter-Stransky et al., 2019*). These TH neurons may act either locally or via projections to regions such as the central amygdala and bed nucleus of the stria terminalis (*Li et al., 2013*; *Matthews et al., 2016*; *Li et al., 2016*; *Groessl et al., 2018*). Many of the TH neurons that participate in these behaviors are located more rostrally, extending into the dorsal raphe (*Taylor et al., 2019*; *Matthews et al., 2016*; *Li et al., 2016*), whereas the recordings in the present study were in the caudal vlPAG. Nevertheless, taken together, the data raise the possibility that anti-nociception, social behavior, fear learning, and/or arousal may also be subject to cerebellar regulation.

Within the caudal vlPAG, stimulation of TH neurons acts primarily through a postsynaptic action of $D_2$ receptors to modulate the synaptic properties of Chx10 neurons; similarly, the anti-nociceptive effects of TH neurons are $D_2$ receptor-mediated (*Meyer et al., 2009*). $D_2$ receptors have been shown to decrease EPSCs through a variety of mechanisms, including disfavoring PKA-dependent

AMPAR phosphorylation (*Håkansson et al., 2006*; *Snyder et al., 2000*; *Shepherd and Huganir, 2007*) or reducing AMPAR surface expression (*Sun et al., 2005*). Likewise, IPSCs can be increased via D$_2$ receptors, but often through presynaptic mechanisms (*Tritsch and Sabatini, 2012*), in contrast to the postsynaptic changes seen here. However, both PKA and PKC can directly modulate GABA receptor conductance (*Browning et al., 1990*; *Kittler and Moss, 2003*), with the direction of modulation depending on the identity of the β-subunit (*Brandon et al., 2002*); phosphorylation of β3-containing GABA receptors leads to increased conductances (*McDonald et al., 1998*).

*Possible functional consequences of the cerebello-vlPAG connection.* The net effect of dopaminergic modulation of PSC strength is a shift in the EI ratio. In cortical circuits, EI ratio is tightly regulated, and plays a role in both circuit function and synaptic plasticity (*Turrigiano and Nelson, 2004*). Dysregulation of EI ratio in mPFC circuitry has been implicated in emotional disorders, including depression and anxiety (*Page and Coutellier, 2019*). Decreasing excitation alone results in an elevated spike threshold; increasing inhibition not only increases spike threshold, but also modulates the gain of the input-output function (*Chance et al., 2002*; *Carvalho and Buonomano, 2009*). Changing the EI ratio in Chx10 cells is therefore likely to change the integrative properties of Chx10 neurons within the freezing circuit through multiple mechanisms.

Nevertheless, if the net inhibitory effect predicted by the shift in EI ratio is the predominant consequence of mCbN activity, then it might account for behavioral data demonstrating reduced freezing following lesions of the cerebellar vermis (*Koutsikou et al., 2014*; *Sacchetti et al., 2002*; *Supple et al., 1988*). Specifically, loss of vermal Purkinje cell activity is predicted to relieve inhibition of the cerebellar nuclei, thereby elevating cerebellar output. The present results suggest that raising mCbN activity could increase signaling by TH cells, elevate D$_2$ receptor activation, and bias Chx10 neurons toward less activity, thus having a suppressive effect on freezing. Conversely, in intact rodents, increases in vermal Purkinje cell activity might relieve dopaminergic tone, thus facilitating, intensifying, or prolonging freezing events, perhaps over a somewhat longer time scale that is commensurate to G-protein coupled signaling. Additionally, if the sex differences in spontaneous firing rates are maintained in vivo, it may result in differences in the dopaminergic tone in the vlPAG, giving rise either to distinct behavioral responses to threatening stimuli or different mechanisms by which common responses are elicited (*Mercer et al., 2016*; *Jain et al., 2019*). Finally, since dopaminergic neurons in the VTA also receive direct input from the cerebellar nuclei (*Carta et al., 2019*), the present study adds to the evidence that the cerebellum may have a substantial role in activating modulatory systems within the brain.

# Materials and methods

**Key resources table**

| Reagent type (species) or resource | Designation | Source or reference | Identifiers | Additional Information |
|---|---|---|---|---|
| Strain, strain background *Mus musculus* | Chx10-cre | Obtained from Jessell Laboratory (*Crone et al., 2008*) | | |
| Strain, strain background *Mus musculus* | tdTomato (B6.Cg-Gt(ROSA) 26Sortm14(CAG-td Tomato)Hze/J) | Jackson Laboratories | Stock: 007914 | |
| Strain, strain background *Mus musculus* | ChR2-EYFP (B6.Cg-*Gt(ROSA)26 Sortm32(CAG-COP4*H134R/EYFP)Hze*/J) | Jackson Laboratories | Stock: 024109 | |
| Strain, strain background *Mus musculus* | GAD2-cre (B6N.Cg-Gad2 tm2(cre)Zjh/J) | Jackson Laboratories | Stock:019022 | |
| Strain, strain background *Mus musculus* | TH-cre (B6.Cg-7630403G23 RikTg(Th-cre)1Tmd/J) | Jackson Laboratories | Stock: 008601 | |

*Continued on next page*

*Continued*

| Reagent type (species) or resource | Designation | Source or reference | Identifiers | Additional Information |
|---|---|---|---|---|
| Strain, strain background *Mus musculus* | L7-Cre (B6.Cg-Tg(Pcp2-cre) 3555Jdhu/J) | Jackson Laboratories | Stock: 010536 | |
| Recombinant DNA reagent | AAV2-EF1α-DIO-hChR2 (H134R)-eYFP | UNC Viral Vector Core | | |
| Recombinant DNA reagent | AAV2-hSyn-hChR2 (H134R)-eYFP-WPRE-PA | UNC Viral Vector Core | | |
| Recombinant DNA reagent | AAVdj-hSyn-hChR2 (E123A)-eYFP-WPRE | Stanford Viral Vector Core | | |
| Recombinant DNA reagent | AAV2-EF1α-DIO-eYFP | UNC Viral Vector Core | | |
| Chemical compound, drug | Cholera Toxin Subunit B (Recombinant), Alexa Fluor 488 Conjugate | ThermoFisher Scientific | Cat. No. C22841 | |
| Chemical compound, drug | Lumafluor red retrobeads | LumaFluor | Red Retrobeads IX | |
| Chemical compound, drug | DNQX | Tocris | Cat. No. 0189 | |
| Chemical compound, drug | CPP | Tocris | Cat. No. 0247 | |
| Chemical compound, drug | CPCCOEt | Tocris | Cat. No. 1028 | |
| Chemical compound, drug | Sulpiride | Tocris | Cat. No. 0894 | |
| Chemical compound, drug | SCH 23390 | Tocris | Cat. No. 0925 | |
| Chemical compound, drug | Prazosin | Tocris | Cat. No. 0623 | |
| Chemical compound, drug | Sotalol | Tocris | Cat. No. 0952 | |
| Chemical compound, drug | Quinpirole | Tocris | Cat. No. 1061 | |
| Chemical compound, drug | Dopamine | Tocris | Cat. No. 3548 | |
| Chemical compound, drug | Isoproterenol | Tocris | Cat. No 17.47 | |
| Software, algorithm | FreezeFrame | Actimetrics (www.actimetrics.com) | | |

## Mice

All procedures conformed to NIH guidelines and were approved by the Northwestern University Institutional Animal Care and Use Committee, protocol IS00000242 (IMR). Mice were housed on a 14:10 light:dark cycle, with *ad lib* access to food and water. The following mice were obtained from Jackson Laboratories: 'Ai14,' which express cre-dependent tdTomato (B6.Cg-*Gt(ROSA)26Sor*$^{tm14}$ $^{(CAG-tdTomato)Hze}$/J, RRID Jax 007914); 'Ai32,' which express cre-dependent ChR2-EYFP (B6.Cg-*Gt (ROSA)26Sor*$^{tm32(CAG-COP4*H134R/EYFP)Hze}$/J, RRID Jax 024109); 'GAD2,' which express cre in GAD2-positive cells (B6N.Cg-*Gad2*$^{tm2(cre)Zjh}$/J, RRID Jax 019022); 'TH,' which express cre in TH-positive cells (B6.Cg-*7630403G23Rik*$^{Tg(Th-cre)1Tmd}$/J, RRID Jax 008601); and 'L7,' which express cre in Purkinje cells (B6.Cg-Tg(Pcp2-cre)3555Jdhu/J, RRID Jax 010536). 'Chx10' mice, which express cre in Chx10 expressing cells, were shared by Dr. Thomas Jessell (*Crone et al., 2008*). All mice were on a C57Bl6/J background. Cre-dependent transgenic lines were maintained as heterozygotes and bred to homozygous Ai14 or Ai27 mice to generate F1 offspring expressing ChR2 with EYFP or tdTomato in the desired cell population. For simplicity, mice are referred to by the cre-expressing promoter and the cre-dependent protein of interest, e.g. Chx10-ChR2 mice. Recordings were made from both

male and female mice and sex was recorded and reported along with n-values. Where sample size permitted, sex was considered as a biological variable in post-hoc analyses.

## Freezing behavior

To stimulate Chx10 neurons in vivo, a fiber optic cannula was implanted unilaterally just above the vlPAG. Stainless steel fiber optic cannulae (200 μm core, 0.39 NA) were cut at a ~ 45° angle to the desired length with a ruby fiber scribe. Mice were fully anesthetized either with isoflurane (2–3%) or ketamine/xylazine injection (80–100 mg/kg ketamine, 5–10 mg/kg xylazine). A craniotomy was made above the PAG, with the medial and posterior coordinates (from bregma) of 0.55–0.75 mm and 4.4–4.75 mm, respectively, adjusted to avoid rupturing the mid-sagittal and transverse sinus. The cannula was lowered to a depth of 2.6 mm and secured with dental cement. After surgery, mice were given 0.015–0.051 mg/kg buprenorphine SR (subcutaneously) and monitored for 3 days.

Mice were placed in a 40 × 40 cm behavioral chamber. The fiber optic cable was connected through a rotary joint to the fiber optic cannula. Video monitoring and light stimuli were controlled by FreezeFrame software (Actimetrics, Wilmette IL). Light trains (10 ms pulses, 50 Hz, 4–6 mW, 465 nm) were applied for 2–5 s through an LED (Doric Lenses). Mice were exposed to 50 consecutive trains, with a start-to-start interval of 20 s. Motion was detected in FreezeFrame by a significant motion pixel algorithm (*Kopec et al., 2007*). The frame rate for comparison of relative motion was 3.75/sec, giving a temporal resolution of 266 ms. After thresholding to identify periods of immobility ('freezing'), the data were further analyzed with Matlab. Motion across each trial of 50 trains was averaged. The percent time of freezing was calculated before ('baseline') and during stimulation ('test'). Baseline was taken as the 2 s period before stimulation. To compare behavior across mice, the z-Score of the motion was calculated. Since the threshold of 3 SDs below the mean had a latency of ~500–600 ms, the test period for evaluating whether freezing had occurred was set as 1 s after light onset and lasted for the length of stimulation.

## Viral and tracer injections

Stereotaxic injections were made with a Patchstar micromanipulator (Scientifica). Viruses and tracers (100–300 nL) were loaded into glass microelectrodes for application by either pressure injection or a Nanoject III (Drummond Scientific). Mice (p28-p35) were anesthetized with isoflurane (2–3%). The exposed scalp was cleaned with 70% ethanol and betadine and locally anesthetized with lidocaine. A craniotomy was made over the targeted brain area and injections were made at (in mm from bregma) −6.23 posterior,±0.6 lateral, −3.3 deep for mCbN or −4.6 posterior,±0.55 lateral, −3.15 deep for vlPAG. For pressure injections,~300 nL of virus or tracer was backfilled into the glass microelectrode and manually injected using 3–5 ms pulses of pressurized oxygen at <20 psi. For the Nanoject III injections, virus or tracer was injected at a rate of 1 nL/sec for 30 s at a time. After injection, the microelectrode was left in place for >1 min before removal, to prevent backflow and allow time for diffusion. The incision was repaired with vetbond and treated with antibiotic ointment. Post-surgical analgesia and monitoring were as above. Mice were allowed to recover for at least 3 days before behavioral testing. The viruses each express a form of ChR2 and EYFP and were the following: AAV2-EF1α-DIO-hChR2(H134R)-eYFP (UNC viral vector core, titer: $4.2 \times 10^{12}$), AAV2-hSyn-hChR2(H134R)-eYFP-WPRE-PA (UNC viral vector core; titer: $5.6 \times 10^{12}$), AAV2-EF1α-DIO-eYFP (UNC viral vector core; titer $4.6 \times 10^{12}$), or AAVdj-hSyn-hChR2(E123A)-eYFP-WPRE (Stanford viral vector core; titer: $6.2 \times 10^{13}$). For retrograde anatomical tracing, red retrobeads (Lumafluor) and CTb-GFP (ThermoFisher) were used.

## Preparation of acute slices

Cerebellar or Mc slices were prepared from p17-p24 mice. Mice were fully anesthetized by isoflurane (2–3%) and transcardially perfused with 10 mL of warm (37°C), oxygenated (95% $O_2$/5% $CO_2$) aCSF, which contained (in mM): 123 NaCl, 3.5 KCl, 1.25 $NaHPO_4$, 26 $NaHCO_3$, 1 $MgCl_2$, 1.5 $CaCl_2$, 10 D-glucose (290–310 mosmol, pH 7.3). Coronal or sagittal slices (250–300 μm) were prepared in warm, oxygenated aCSF to facilitate cutting through heavy myelination (*Person and Raman, 2012*; *Wu and Raman, 2017*). PAG slices were cut from p21-p80 mice. Mice were perfused with 10 mL of cold (4°C), oxygenated sucrose cutting solution, which contained (mM): 83 NaCl, 2.5 KCl, 1 $NaH_2PO_4$, 26.2 $NaHCO_3$, 22 dextrose, 72 sucrose, 0.5 $CaCl_2$, and 3.3 $MgCl_2$, one kynurenate (300–

310 mosmol, pH 7.3). Coronal slices (300 µm) were cut in cold sucrose cutting solution. Both cerebellar and PAG slices recovered in oxygenated aCSF for 30–60 min at 37°C and then were maintained at room temperature (22°−23°C) until use.

## Electrophysiological recording

Whole cell voltage- and current-clamp recordings were made from neurons in the mCbN, the vlPAG, and Mc. The extracellular solution contained (mM) 123 NaCl, 3.5 KCl, 1.25 NaHPO$_4$, 26 NaHCO$_3$, 1 MgCl$_2$, 1.5 CaCl$_2$, 10 D-glucose (290–310 mosm, pH 7.3); for recordings with synaptic stimulation, the Ca concentration was increased to 2 mM to increase release probability. Voltage clamp recordings were made with one of two intracellular solutions: Cs-gluconate, which contained (mM): 120 CsCH$_3$SO$_3$, 3 NaCl, 2 MgCl$_2$, 1 EGTA, 10 HEPES, 4 MgATP, 0.3 Tris-GTP, 14 Tris-creatine phosphate, 1.2 QX-314, 4 TEA-Cl, 12 sucrose (288 mOsm, buffered with CsOH to pH 7.32) or K-gluconate, which contained (in mM): 130 K-gluconate, 2 Na-gluconate, 6 NaCl, 2 MgCl$_2$, 0.1 CaCl$_2$, 1 EGTA, 4 MgATP, 0.3 TrisGTP, 14 Tris-creatine phosphate, 10 sucrose, 10 HEPES, 5 QX-314 (287 mOsm, buffered with KOH to pH 7.35). Current-clamp recordings were made with the K-gluconate internal solution, without QX-314.

Borosilicate patch pipettes were pulled to 2–5 MΩ on a Sutter P-97 puller. The liquid junction potential was −7 mV; for exact voltages, 7 mV should be subtracted from values in the text and figures. Temperature was maintained at 35–37°C by a Warner TC-324B controller. Data were digitized at 20–50 kHz and filtered at 10 kHz, acquired with a Multiclamp 700B amplifier, Digidata 1440A converter, and Clampex 10 software. During voltage-clamp recordings, access resistance was monitored by a 10 mV hyperpolarization before each sweep, and recordings with changes > 30% were discarded. Access resistances ranged from 1.1 to 35.8 MΩ The average access resistance was 7.2 ± 0.3 MΩ for mCbN neurons, 16.3 ± 2.0 MΩ for Chx10 neurons, and 13.7 ± 0.8 MΩ for TH cells. During current clamp, the bridge was balanced. Cells were identified under DIC optics with a Scientifica Sci-Cam Pro camera and Ocular software package. Fluorescently labeled neurons were identified under illumination with a ThorLabs LED (530 nm).

Recordings were made from mCbN neurons with cell bodies > 20 µm in diameter. In the mCbN, these large neurons include both glutamatergic and glycinergic projection neurons, whose intrinsic properties show no significant differences (*Bagnall et al., 2009*). Spontaneous firing rates were recorded with no injected current. Frequency-intensity (FI) curves were made with 500 ms current steps (−300 and 300 pA, 25 pA steps). IPSCs were evoked in mCbN cells either by applying 2 ms light pulses (470 nm) through the objective in slices from L7-ChR2 mice (*Najac and Raman, 2015*), or by positioning a parallel bipolar electrode within the mCbN or in the white matter just dorsal to the mCbN and stimulating electrically with 0.1 ms, 0.7–1.0 mA pulses through a stimulus isolation unity (Warner Instruments), controlled by a Master-8 (AMPI).

In the vlPAG, targeted recordings were made from labeled neurons in Chx10-tdT or TH-tdT mice. Spontaneous recordings from Chx10 and TH neurons were made with no injected current. FI curves in Chx10 cells were made with 500 ms current steps (−100 to 100 pA, 10 pA steps). EPSCs and IPSCs were evoked electrically as above, with a concentric or parallel bipolar stimulating electrode placed within the vlPAG, with stimulus intervals of 10–20 s with ≥10 sweeps per condition. EPSCs and IPSCs were isolated by recording at −70 mV (near E$_{Cl}$) and 0 mV (near E$_{cation}$) respectively. To stimulate ChR2-expressing mCbN axons, 30 full-field light pulses (2–5 ms, 470 nm) were applied at 25 Hz with a ThorLabs LED (max power through the objective, 4.7 mW). In the Mc, large neurons were held at −70 mV, and EPSCs were evoked optogenetically as in the PAG, with two pulses with a 40 ms interval.

All drugs were from Tocris Biosciences and were bath applied where indicated at the following concentrations: 10 µM DNQX, 10 µM CPP, 20 µM CPCCOEt, 1 µM sulpiride, 1 µM SCH-23390, 5 µM prazosin, 30 µM sotalol, 25 µM quinpirole, 10 µM dopamine, 5 µM isoproterenol.

## Data Analysis

Electrophysiological data were analyzed with AxographX and IPro 7.08 (Wavemetrics). Action potentials were detected and their waveforms analyzed in AxographX. Phase-plane plots were generated in IgorPro from the time derivative of the membrane potential. Action potential threshold was defined as the membrane potential at which dV/dt exceeded 10 mV/ms. Peak EPSCs and IPSCs

were measured from the baseline-zeroed average of $\geq$10 traces. IPSCs decay phases were fit with the sum of two exponentials, and weighted time constants were calculated from the percent contribution of each component to the peak current.

### Image acquisition and processing

Mice were anesthetized with 60–100 mg/kg Na-pentobarbital and transcardially perfused with 10 mL 0.1 M PBS followed by 10 mL 4% formaldehyde in 0.1 M PBS. Brains were removed and post-fixed overnight in 4% formaldehyde (room temperature). Sections (50–100 µm) were cut on a Leica 1000S microtome and mounted on glass slides. Confocal images were acquired with a Leica SP5 laser scanning confocal microscope in the Northwestern University Biological Imaging Facility. Images were processed with open-source FIJI software (*Schindelin et al., 2012*). Images were adjusted for brightness and contrast. For images of axonal arborization, the black and white image was color-inverted for visual clarity.

### Statistics

Data are reported as mean ± S.E.M. Statistical tests were performed in Excel and GraphPad Prism. Statistics were calculated with two-sample paired or unpaired Student's t-tests, one-sample t-tests for normalized data, or a Kolmogorov-Smirnov test, as indicated in the text. Significance was taken as $p < 0.05$, and *p*-values are reported. The n values refer to the number of cells recorded or mice tested as indicated (e.g., n = x cells); values in brackets indicate the number of observations in each sex (M, male, F, female).

## Acknowledgements

We thank the members of the Raman lab for helpful discussions and Dr. Thomas Jessell for the Chx10-cre mice. We thank former lab member Dr. Audrey Mercer for the recordings from interpositus nucleus. Confocal imaging was performed at the Biological Imaging Facility at Northwestern University.

## Additional information

### Funding

| Funder | Grant reference number | Author |
| --- | --- | --- |
| National Institute of Neurological Disorders and Stroke | F32 NS106720 | Christopher E Vaaga |
| National Institute of Neurological Disorders and Stroke | R37 NS39395 | Indira M Raman |

The funders had no role in study design, data collection and interpretation, or the decision to submit the work for publication.

### Author contributions

Christopher E Vaaga, Conceptualization, Funding acquisition, Investigation, Visualization, Writing - original draft, Writing - review and editing, Conducted all electrophysiological, anatomical, and behavioral experiments, experimental design and analysis; Spencer T Brown, Investigation, Methodology, Helped with all aspects of experimental design, execution, and analysis of behavioral experiments; Indira M Raman, Conceptualization, Supervision, Funding acquisition, Visualization, Writing - review and editing, experimental design and analysis

### Author ORCIDs

Christopher E Vaaga (iD) https://orcid.org/0000-0001-9777-3808
Indira M Raman (iD) https://orcid.org/0000-0001-5245-8177

## Ethics

Animal experimentation: All procedures conformed to NIH guidelines and were approved by the Northwestern University Institutional Animal Care and Use Committee, protocol IS00000242 (IMR).

## Decision letter and Author response

Decision letter https://doi.org/10.7554/eLife.54302.sa1
Author response https://doi.org/10.7554/eLife.54302.sa2

# Additional files

## Supplementary files

• Transparent reporting form

## Data availability

All data generated during this study are included in the manuscript and supporting files, and values of individual measurements within a population are included in graphs of data.

The following datasets were generated:

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
