## [Decision Letter]

**Acceptance summary:**

The role of the cerbellum in the regulation of the innate rodent defensive behavior, freezing, has not previously been explored. This paper represents a compelling characterization of the circuitry in the ventrolateral periaqueductal gray (vlPAG) known to produce freezing behavior, and implicates the cerebellum as one controlling input. The work also suggests that the dopaminergic vlPAG neurons are a major control node for defensive freezing, and that inhibiting the cerebellar afferents to these cells promotes freezing.

**Decision letter after peer review:**

Thank you for submitting your article "Cerebellar modulation of synaptic input to freezing-related neurons in the periaqueductal gray" for consideration by *eLife*. Your article has been reviewed by three peer reviewers, including Julie A Kauer as the Reviewing Editor and Reviewer #1, and the evaluation has been overseen by a Reviewing Editor and Richard Aldrich as the Senior Editor. The following individuals involved in review of your submission have agreed to reveal their identity: Philip Tovote (Reviewer #3).

The reviewers have discussed the reviews with one another and the Reviewing Editor has drafted this decision to help you prepare a revised submission.

Summary:

This manuscript provides evidence for the idea that medial cerebellar nucleus cells excite dopaminergic PAG cells that in turn inhibit Chx10 cells that can produce freezing, using optogenetic and electrophysiological analysis. The work represents a compelling characterization of the vlPAG circuitry known to produce freezing behavior, and implicates the cerebellum as (at least) one controlling afferent. Moreover, this work suggests one key role for the dopaminergic vlPAG neurons. The experiments are carried out to a high standard and the results are credible and of interest to the field.

Essential revisions:

The reviewers were generally very positive about the paper, but feel that more data are essential to validate the major conclusions:

1) The sample size for the behavioral experiments is quite low (n = 4 ChR2, n = 2 control). For experiments in freely moving mice, the N needs to be increased. Even though the effects seem very robust, at least 5-6 mice per group should be expected to avoid statistical errors. If the data are analyzed by sex, any differences should be noted, given the electrophysiological dependence on sex.

2) The authors indicate a very interesting requirement for D2R mediated modulation of the circuit, but do not look further at whether a D2R antagonist prevents modulation by optical stimulation of mCB fibers. Please add these data.

3) Dr. Susan Ingram has done a number of experiments demonstrating that dopamine signaling in the PAG can regulate behavior, the authors should discuss this. In addition, there have been several papers in the last several years focusing on DA neurons in this region and regulation of behavior, from the labs of Tye (Social behavior), Haubensak (fear learning), Kash (pain), Brown (Pain) and Weinshenker (arousal). Given the important role for vlPAG DA activation by the cerebellum, they should be discussed more fully.

4) Anatomical placements of optical fibers need to be reported.

5) Some more information as to the nature of the evoked IPSCs is needed to allow interpretation of synaptic connectivity between vlPAG neuronal subtypes better. For example, in Figure 5A,B, mCbN light stimulation evoked EPSCs in unidentified, Chx10 and GAD2 neurons at low probability but no IPSCs were detected in the same cells. The authors show that mCbN neurons target TH^+^ neurons in vlPAG at higher probability and excitation of these neurons results in the enhancement of IPSCs amplitude in Chx10 neurons through a D_2_-dependent mechanisr. These results suggest that stimulation of mCbN neurons could evoke polysynaptic IPSCs in Chx10 neurons. Have you observed IPSCs in no-EPSCs cells? In addition, it would be interesting to know how Chx10 neurons response to each light stimulation pulses to ChR2 positive TH^+^ neurons.

---

## [Author Response]

Summary:This manuscript provides evidence for the idea that medial cerebellar nucleus cells excite dopaminergic PAG cells that in turn inhibit Chx10 cells that can produce freezing, using optogenetic and electrophysiological analysis. The work represents a compelling characterization of the vlPAG circuitry known to produce freezing behavior, and implicates the cerebellum as (at least) one controlling afferent. Moreover, this work suggests one key role for the dopaminergic vlPAG neurons. The experiments are carried out to a high standard and the results are credible and of interest to the field.

Thank you for this assessment of our work.

Essential revisions:The reviewers were generally very positive about the paper, but feel that more data are essential to validate the major conclusions:1) The sample size for the behavioral experiments is quite low (n = 4 ChR2, n = 2 control). For experiments in freely moving mice, the N needs to be increased. Even though the effects seem very robust, at least 5-6 mice per group should be expected to avoid statistical errors. If the data are analyzed by sex, any differences should be noted, given the electrophysiological dependence on sex.

We have increased the number of mice used for behavioral experiments. There are now 7 control mice (which did not change their freezing behavior with light stimuli) and 7 Chx10-ChR2 mice, which significantly froze. Results were robust and indistinguishable in both sexes. Owing to limitations in animal numbers, we did not aim for statistical comparisons of sex differences. The new data is included in Figure 2 and in subsection “Optogenetic stimulation of vlPAG Chx10 neurons in vivo”.

2) The authors indicate a very interesting requirement for D2R mediated modulation of the circuit, but do not look further at whether a D2R antagonist prevents modulation by optical stimulation of mCB fibers. Please add these data.

These data were in fact included in the original manuscript in Figure 8E-H. We agree that this experiment should be placed more prominently so that it will not be overlooked. We have now separated the data so that the new Figure 8 includes the pharmacological identification of dopamine receptor subtype and the new Figure 9 demonstrates the blockade of mCbN-dependent modulation of PSCs by D_2_ receptor antagonists. The text (subsection “Mechanism of mCbN-induced modulation of postsynaptic currents”) has been modified accordingly.

3) Dr. Susan Ingram has done a number of experiments demonstrating that dopamine signaling in the PAG can regulate behavior, the authors should discuss this. In addition, there have been several papers in the last several years focusing on DA neurons in this region and regulation of behavior, from the labs of Tye (Social behavior), Haubensak (fear learning), Kash (pain), Brown (Pain) and Weinshenker (arousal). Given the important role for vlPAG DA activation by the cerebellum, they should be discussed more fully.

Yes, these are exciting papers that raise the question of whether the behavioral effects of the cerebello-PAG connection may be quite extensive. Initially, we did not include much speculation on these behaviors because we did not wish to overstate the present results, which focus on the discoveries of the microcircuitry, synaptic properties, and dopaminergic modulation; behaviorally, we only examine effects of Chx10 activation rather than modulation of behavior by dopamine. Also, many of these studies focus on TH neurons in the dorsal raphe proper, which is more rostral (and ventral to the central aqueduct) than the region studied here. Nevertheless, we have now included references to work from the labs as suggested and raised the question of whether those behaviors might be subject to cerebellar regulation (subsection “Effects of mCbN input on neurons of the vlPAG”).

4) Anatomical placements of optical fibers need to be reported.

The anatomical placements are now reported in the manuscript in subsection “Optogenetic stimulation of vlPAG Chx10 neurons in vivo”. These measurements confirm that the low-magnification transmitted light image of the fiber track in Figure 2D is representative of the rest of the data.

5) Some more information as to the nature of the evoked IPSCs is needed to allow interpretation of synaptic connectivity between vlPAG neuronal subtypes better. For example, in Figure 5A,B, mCbN light stimulation evoked EPSCs in unidentified, Chx10 and GAD2 neurons at low probability but no IPSCs were detected in the same cells. The authors show that mCbN neurons target TH^+^ neurons in vlPAG at higher probability and excitation of these neurons results in the enhancement of IPSCs amplitude in Chx10 neurons through a D2-dependent mechanisr. These results suggest that stimulation of mCbN neurons could evoke polysynaptic IPSCs in Chx10 neurons. Have you observed IPSCs in no-EPSCs cells? In addition, it would be interesting to know how Chx10 neurons response to each light stimulation pulses to ChR2 positive TH^+^ neurons.

Yes, the idea of mCbN-evoked polysynaptic IPSCs onto Chx10 neurons is an interesting and reasonable possibility, but the data provide evidence against it, which we have now clarified in the manuscript. First, mCbN stimulation never evoked IPSCs (N=42/42 cells), regardless of whether an mCbN-evoked EPSC was evident, suggesting that cerebellar input to GAD2 cells is not strong enough to elicit polysynaptic IPSCs in Chx10 cells. This information is now stated explicitly in subsection “Effects of mCbN input to the vlPAG”. The recruitment of additional inhibitory inputs as a mechanism for the modulation of IPSC is also inconsistent with the finding that intracellular application of GDP-βS blocks the mCbN induced increase in IPSC amplitude, which suggests a postsynaptic site of action. This clarification has been added to subsection “Mechanism of mCbN-induced modulation of postsynaptic currents”. Finally, regarding the responses to trains of light pulses to TH neurons, no evoked EPSCs or IPSCs were evident (currents at times of expected PSCs were <5 pA), suggesting that co-release is unlikely within the PAG (unlike in the CeA, Groessl et al., 2018). The lack of response to the train stimulation of TH cells has been added to Figure 7, and the text (subsection “Mechanism of mCbN-induced modulation of postsynaptic currents”).